# Chemoproteomics validates selective targeting of *Plasmodium* M1 alanyl aminopeptidase as an antimalarial strategy

Carlo Giannangelo[1†], Matthew P Challis[1†], Ghizal Siddiqui[1], Rebecca Edgar[2,3], Tess R Malcolm[4,5], Chaille T Webb[4,5], Nyssa Drinkwater[4,5], Natalie Vinh[6], Christopher Macraild[1], Natalie Counihan[2,3], Sandra Duffy[7], Sergio Wittlin[8,9], Shane M Devine[10,11], Vicky M Avery[7,12], Tania De Koning-Ward[2,3], Peter Scammells[6], Sheena McGowan[4,5], Darren J Creek[1]*

[1]Drug Delivery, Disposition and Dynamics, Monash Institute of Pharmaceutical Sciences, Monash University, Parkville, Australia; [2]School of Medicine, Deakin University, Geelong, Australia; [3]The Institute for Mental and Physical Health and Clinical Translation, Deakin University, Geelong, Australia; [4]Monash Biomedicine Discovery Institute and Department of Microbiology, Monash University, Clayton, Australia; [5]Centre to Impact AMR, Monash University, Clayton, Australia; [6]Medicinal Chemistry, Monash Institute of Pharmaceutical Sciences, Monash University, Parkville, Australia; [7]Discovery Biology, Centre for Cellular Phenomics, Griffith University, Nathan, Australia; [8]Swiss Tropical and Public Health Institute, Allschwil, Switzerland; [9]University of Basel, Basel, Switzerland; [10]The Walter and Eliza Hall Institute of Medical Research, Parkville, Australia; [11]Department of Medical Biology, The University of Melbourne, Parkville, Australia; [12]School of Environment and Science, Griffith University, Nathan, Australia

*For correspondence:
darren.creek@monash.edu

†These authors contributed equally to this work

Competing interest: The authors declare that no competing interests exist.

**Abstract** New antimalarial drug candidates that act via novel mechanisms are urgently needed to combat malaria drug resistance. Here, we describe the multi-omic chemical validation of *Plasmodium* M1 alanyl metalloaminopeptidase as an attractive drug target using the selective inhibitor, MIPS2673. MIPS2673 demonstrated potent inhibition of recombinant *Plasmodium falciparum* (*Pf*A-M1) and *Plasmodium vivax* (*Pv*A-M1) M1 metalloaminopeptidases, with selectivity over other *Plasmodium* and human aminopeptidases, and displayed excellent in vitro antimalarial activity with no significant host cytotoxicity. Orthogonal label-free chemoproteomic methods based on thermal stability and limited proteolysis of whole parasite lysates revealed that MIPS2673 solely targets *Pf*A-M1 in parasites, with limited proteolysis also enabling estimation of the binding site on *Pf*A-M1 to within ~5 Å of that determined by X-ray crystallography. Finally, functional investigation by untargeted metabolomics demonstrated that MIPS2673 inhibits the key role of *Pf*A-M1 in haemoglobin digestion. Combined, our unbiased multi-omic target deconvolution methods confirmed the on-target activity of MIPS2673, and validated selective inhibition of M1 alanyl metalloaminopeptidase as a promising antimalarial strategy.

## eLife assessment

The manuscript makes an **important** contribution to antimalarial drug discovery, utilizing diverse systems biology methodologies. It focuses on an improved M1 metalloprotease inhibitor and

provides **compelling** evidence for the utility of chemoproteomics in pinpointing PfA-M1 targeting. Additionally, metabolomic analysis reveals specific alterations in the final steps of hemoglobin breakdown. These findings highlight the potential of the developed methodology not only for PfA-M1 targeting but also for other inhibitors targeting various malarial proteins or pathways.

## Introduction

Malaria is a devastating parasitic disease that causes major health, economic, and societal impacts to almost half of the world's population that live in malaria-endemic countries. In 2021 there were an estimated 247 million cases of malaria globally and 619,000 associated deaths (*WHO, 2022*). An estimated 95% of malaria deaths are attributed to *P. falciparum*, the species responsible for the most severe and lethal form of the disease (*WHO, 2022*). *Plasmodium vivax* is also responsible for significant morbidity, and is the most geographically widespread malaria parasite (*Price et al., 2020*). Although the global incidence rate of malaria has declined over the last decade, the number of people worldwide suffering and dying from malaria is now increasing and parasite resistance has emerged to all current antimalarial drug classes, including the frontline artemisinin therapies (*WHO, 2022*; *Phillips et al., 2017*). Alarmingly, artemisinin-based combination therapies fail to cure infections in up to 50% of patients in some regions of Asia (*van der Pluijm et al., 2019*) and delayed parasite clearance rates following artemisinin-based therapy (synonymous with decreased parasite susceptibility to artemisinin) has now been detected in Africa (*Balikagala et al., 2021*), where most malaria deaths occur. This, combined with the lack of mechanistic diversity in emerging antimalarials in development, means there is an urgent need for new drug classes with novel mechanisms of action that bypass resistance and treat both *P. falciparum* and *P. vivax* infections.

Haemoglobin (Hb) digestion is an essential metabolic process and a point of key vulnerability during *Plasmodium* parasite intraerythrocytic development. The parasite digests up to 75% of host cell Hb to provide amino acids for protein synthesis and this process additionally serves non-anabolic functions, such as maintaining osmotic stability of the infected red blood cell (*Liu et al., 2006*; *Lew et al., 2003*; *Krugliak et al., 2002*; *Loria et al., 1999*). *Plasmodium* parasites employ the interplay of numerous proteases from different classes to digest Hb into its constituent amino acids (*Goldberg, 2005*). *P. falciparum* M1 alanyl aminopeptidase (PfA-M1) and M17 leucyl aminopeptidase (PfA-M17) are two zinc-dependent metalloaminopeptidases involved in the terminal stage of Hb digestion (*Edgar et al., 2022*; *Dalal and Klemba, 2007*). Both PfA-M1 and PfA-M17 are thought to be essential for parasite survival and are promising targets for antimalarial therapy in vitro and in vivo (*Edgar et al., 2022*; *Dalal and Klemba, 2007*; *Skinner Adams et al., 2007*; *Skinner Adams et al., 2012*; *Harbut et al., 2011*; *McGowan, 2013*; *Nankya-Kitaka et al., 1998*; *Bounaadja et al., 2017*; *Deprez-Poulain et al., 2012*; *Drinkwater et al., 2016b*; *Vinh et al., 2019*; *González-Bacerio et al., 2014*; *Mistry et al., 2014*; *Ruggeri et al., 2015*; *Kannan Sivaraman et al., 2013*). However, attempts at genetic manipulation of PfA-M1 have been unsuccessful to date, which has precluded formal genetic validation of PfA-M1 as a drug target. Bestatin-based inhibitors of PfA-M1 were previously shown to prevent proteolysis of Hb peptides, leading to a swollen digestive vacuole and stalled growth at the trophozoite stage (*Harbut et al., 2011*). Loss of PfA-M17 function through chemical inhibition or conditional knockdown also results in stalled growth at the trophozoite stage but does not induce digestive vacuole swelling (*Edgar et al., 2022*). Instead, specific loss of PfA-M17 function results in the presence of multiple membrane-bound digestive vacuoles per parasite. We have previously used rational drug design to optimise dual M1 and M17 inhibitors with nanomolar potency across multiple *Plasmodium* species (*Drinkwater et al., 2016b*; *Vinh et al., 2019*; *Mistry et al., 2014*; *Kannan Sivaraman et al., 2013*). More recently, we reported a selective PfA-M17 inhibitor with potent antimalarial activity (*Edgar et al., 2022*).

Like PfA-M17, PfA-M1 is an attractive target for novel antimalarial drug development. The PfA-M1 enzyme is a monomeric protein consisting of four domains and is widely believed to function in the cytosol, although localisation to other compartments has been reported (*Dalal and Klemba, 2007*; *Harbut et al., 2011*; *Allary et al., 2002*; *Azimzadeh et al., 2010*; *Mathew et al., 2021*). It has a broad substrate specificity that includes most basic and hydrophobic amino acids, but preferentially cleaves leucine and methionine (*Allary et al., 2002*; *Poreba et al., 2012*). PfA-M1 uses a zinc-dependent mechanism to catalyse the release of single amino acids from the N-terminus of short peptides. All

reported inhibitors of *Pf*A-M1 target the enzyme active site and act to chelate the catalytic zinc ion (*Skinner Adams et al., 2007*; *Harbut et al., 2011*; *Deprez-Poulain et al., 2012*; *Drinkwater et al., 2016b*; *Vinh et al., 2019*; *Poreba et al., 2012*; *McGowan et al., 2009*; *Flipo et al., 2007*; *Harbut et al., 2008*; *Velmourougane et al., 2011*). Obtaining selectivity for *Pf*A-M1 over *Pf*A-M17 is challenging as the enzymes share similar architecture within the active sites and a catalytic mechanism involving zinc ions (*McGowan et al., 2009*; *McGowan et al., 2010*). As such, very few compounds that selectively inhibit *Pf*A-M1 over *Pf*A-M17 have been reported (*Harbut et al., 2011*; *Bounaadja et al., 2017*; *Flipo et al., 2007*) and unbiased identification of the target in parasites is critical to confirm specificity.

Identifying the molecular targets of compounds is a significant challenge for antimalarial drug development (*Challis et al., 2022*; *Bailey et al., 2023*). One of the most successful approaches to target deconvolution has been in vitro evolution and whole genome analysis (IVIEWGA) (*Carolino and Winzeler, 2020*; *Baragaña et al., 2015*; *Summers et al., 2022*; *Cowell et al., 2018*; *Flannery et al., 2015*). However, IVIEWGA can also point to genes encoding general drug resistance mechanisms rather than the cellular target, and a major goal of new drug development for malaria is the identification of compounds that fail to yield resistant parasites in vitro (irresistible compounds), as these compounds are less likely to result in clinical resistance (*Yang et al., 2021*). Chemoproteomics methods based on mass spectrometry offer alternative routes to validate the molecular target(s) of compounds in parasites. One strategy relies on the use of chemical probes to enrich and identify protein targets (*Siddiqui et al., 2022b*). However, this method requires chemical modification of the drug, which could alter its pharmacological properties. New thermal stability (*Molina et al., 2013*; *Dziekan et al., 2019*) and limited proteolysis (*Piazza et al., 2020*) techniques are two examples of unbiased chemoproteomics methods that provide direct evidence of on-target engagement on a proteome-wide scale, without the need for modification of the drug. These methods rely on the concepts of thermal and proteolytic stabilisation of the protein target due to ligand binding and leverage recent advances in mass spectrometry to detect drug-specific changes to protein or peptide abundance within complex proteomes. Thermal proteome profiling was recently applied in *P. falciparum* to successfully identify the targets of several antimalarial compounds (*Dziekan et al., 2019*; *Wirjanata et al., 2021*). Whereas limited proteolysis proteomics has not previously been reported to identify drug targets in *P. falciparum*, this approach has successfully mapped protein-small molecule interactions in other complex eukaryotic proteomes (*Piazza et al., 2020*).

Here, we report an M1 selective aminopeptidase inhibitor with a hydroxamic acid zinc binding group (*Drinkwater et al., 2016b*) (MIPS2673), which targets both the *P. falciparum* (*Pf*A-M1) and *P. vivax* (*Pv*A-M1) M1 homologues and demonstrates antimalarial activity against a panel of drug-resistant *P. falciparum* strains. We developed orthogonal chemoproteomics approaches based on thermal stability and limited proteolysis, combined with metabolomics analysis, to validate the on-target activity of MIPS2673 in parasites. Our comparatively simple and inexpensive chemoproteomics methods for identifying antimalarial drug targets utilise an efficient data-independent acquisition mass spectrometry (DIA-MS) approach that allows comprehensive and reproducible proteome coverage. Together, our studies confirm *Plasmodium* M1 alanyl aminopeptidase as the target of MIPS2673 and validate selective inhibition of this enzyme as a potential multi-stage and cross-species antimalarial strategy.

## Results
### Design and synthesis of a cross-species hydroxamic acid-based inhibitor selective for *Plasmodium* M1 alanyl aminopeptidase

We previously reported a series of *Pf*A-M1 and M17 aminopeptidase inhibitors that possessed a hydroxamic acid zinc binding group to coordinate catalytic zinc ion(s) and a variety of hydrophobic groups to probe the S1′ binding pocket of these enzymes (*Vinh et al., 2019*). Whilst a number of these modifications successfully improved inhibitory potency, their incorporation reduced polarity and aqueous solubility. One such example is the pivaloyl group present in compound **1** (*Figure 1A*; *Drinkwater et al., 2016b*; *Vinh et al., 2019*). In order to balance these opposing factors, one of the methyl groups of the pivaloyl was replaced by an alcohol to increase polarity whilst maintaining the capacity to engage the S1′ pocket via hydrophobic interactions (*Figure 1A*). This modification resulted in MIPS2673 (compound **4**), which has a significantly lower cLogP (1.45 vs 2.23 for **1**) and

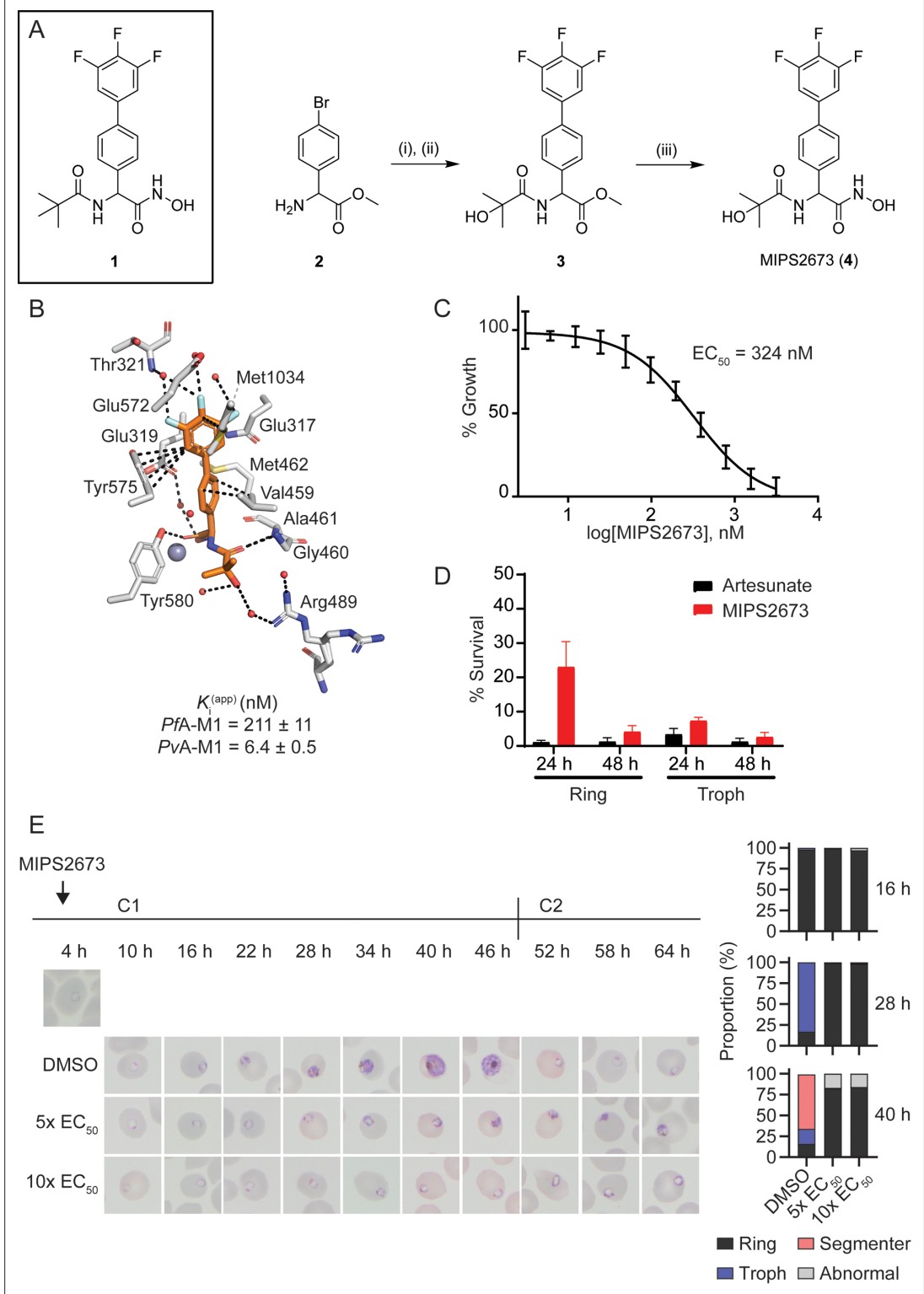

**Figure 1.** Synthesis and antimalarial activity of the M1-selective aminopeptidase inhibitor, MIPS2673. (**A**) Synthesis of MIPS2673 (**4**). *Reagents and conditions*: (**i**) RCOOH, EDCI, DMAP, $CH_2Cl_2$; (**ii**) 3,4,5-trifluorophenylboronic acid, $Pd(PPh_3)_2Cl_2$, $Na_2CO_3$, THF; (**iii**) $NH_2OH \bullet HCl$, KOH. (**B**) Binding mode of MIPS2673 bound to *Pf*A-M1 determined by X-ray crystallography (8SLO.pdb). Interactions between MIPS2673 (orange sticks) and *Pf*A-M1 (grey) shown as black dashed lines. Inhibition constant ($Ki$) for MIPS2673 towards recombinant, purified *Pf*A-M1 and *Pv*A-M1. (**C**) Sensitivity to MIPS2673 exposure (72 hr) for *P. falciparum* 3D7 parasites. Parasite growth was determined relative to a no-drug control by SYBR Green I assay. The $EC_{50}$ value was calculated from four biological replicates performed in technical triplicate and the data plotted as the mean ± standard error of the mean. (**D**) Stage-

*Figure 1 continued on next page*

*Figure 1 continued*

dependent activity of MIPS2673 in *P. falciparum* (3D7). Synchronised ring- or trophozoite-stage cultures were incubated for either 24 hr or 48 hr with compound at 10× $EC_{50}$, before the drug was washed off and parasites allowed to grow for a further 48 hr. Growth was determined via SYBR Green I assay and compared to vehicle (DMSO)-treated controls. Shown is the mean ± standard deviation (n=4). (**E**) MIPS2673-induced morphology and growth effects for *P. falciparum* 3D7 parasites (left). Synchronised cultures at 4 hr post invasion were treated over two cycles (C1, cycle 1; C2, cycle 2) with either 5× or 10× $EC_{50}$ or DMSO at the concentration present in the 10× $EC_{50}$ treatment. Representative Giemsa-stained smears from two biological replicates show parasites stalling and dying at the ring to trophozoite transition. The percentage of parasites at each stage of development is shown from counts of 100 infected red blood cells at 16 hr, 28 hr, and 40 hr post invasion (right).

The online version of this article includes the following source data and figure supplement(s) for figure 1:

**Source data 1.** Data collection and refinement statistics for X-ray crystal structure.

**Figure supplement 1.** Antimalarial potency of the clinically used artemisinin derivative, artesunate.

favourable solubility (i.e. the kinetic solubility of MIPS2673 was estimated to be in the 50–100 μg/mL range using nephelometry vs 12.5–25 μg/mL for **1**). MIPS2673 (**4**) was synthesised from methyl 2-amino-2-(4-bromophenyl)acetate (**2**) in three synthetic steps as shown in ***Figure 1***. Amide formation was achieved via an EDCI-mediated coupling and the trifluorophenyl moiety was installed using a Suzuki-Miyaura coupling to form the synthetic intermediate **3**. Finally, the methyl ester was converted to the corresponding hydroxamic acid using 5 M methanolic KOH and $NH_2OH \cdot HCl$ to afford MIPS2673 (**4**).

We determined the inhibitory activity of MIPS2673 against M1 and M17 enzymes from both *P. falciparum* and *P. vivax* (***Figure 1B***). The binding affinities ($Ki$) of MIPS2673 towards purified, recombinant *Pf*A-M1 show the compound to be a potent inhibitor ($Ki$ = 211 ± 11 nM), with >4-fold selectivity over *Pf*A-M17 ($Ki$ = 921 ± 69 nM). MIPS2673 was significantly more potent against *Pv*A-M1 ($Ki$ = 6.4 ± 0.5 nM) with >155-fold selectivity over *Pv*A-M17 ($Ki$ >1000 nM). We confirmed that MIPS2673 is non-inhibitory against the *Pf*A-M18 ($Ki$ >500 μM) and *Pf*APP ($Ki$ >40 μM) recombinant metalloaminopeptidases (data not shown).

In contrast to the M1 selectivity demonstrated by MIPS2673, the inhibitor **1** was twofold more selective for *Pf*A-M17 vs *Pf*A-M1 (***Vinh et al., 2019***). When bound to *Pf*A-M1, the position of the biaryl of MIPS2673 compares with that of our previous inhibitors (***Drinkwater et al., 2016b***; ***Vinh et al., 2019***). The moiety makes similar interactions with Met1034 (hydrophobic) and Glu319 (carbonyl-pi) (***Figure 1B***). The fluoro-substituents sit deep in the S1 pocket and form the same intricate network of water-mediated hydrogen bonds as **1**, in which the fluorine atoms act as H-bond acceptors. The substitution on the *tert*-butyl, replacing a methyl with an alcohol, is difficult to definitively position in the structure due to the rotation around the C-03 bond. It is possible that this alcohol could form a long H-bond to Arg-489, which should provide some improvement in position and potency. The Arg-489 side-chain shows two orientations in the electron density, suggesting that some rotation of the alcohol is occurring (***Figure 1B***). Inspection of the structure of *Pf*A-M17 with **1** (PDB ID 4ZY2) (***Drinkwater et al., 2016b***) does not reveal why the compound is selective. The *tert*-butyl of **1** is sitting in an open pocket and the introduction of an alcohol is unlikely to induce any clashes, steric or electrostatic.

**Table 1.** Percent inhibition by MIPS2673 of *Pf*A-M1 and *Pv*A-M1 aminopeptidase activities compared to selected human M1 homologues.

| Concentration (μM) | % inhibition* | | | | |
| --- | --- | --- | --- | --- | --- |
| | *Pf*A-M1 | *Pv*A-M1 | LTA4H | ERAP1 | ERAP2 |
| 1.25 | 100 | 100 | 0 | 0 | 31 |
| 10 | 100 | 100 | 58 | 2.3 | 0 |
| 500 | – | – | 42.2 | 89.8 | 97.8 |
| 1000 | – | – | 53.4 | 95.6 | 98.7 |

*Values represent the mean of three independent experiments. All assays were performed at pH 8.0. The concentration of substrate H-Leu-NHMec (H-Leu-NHMec, L-leucine-7-amido-4-methylcoumarin hydrochloride) varied depending on the enzyme assayed 20 μM for *Pf*A-M1, 40 μM for *Pv*A-M1, and 100 μM for ERAP1 and ERAP2. LTA4H was assessed using the substrate H-Ala-NHMec (L-alanine-7-amido-4-methylcoumarin hydrochloride) at a concentration of 20 μM. LTA4H, leukotriene A-4 hydrolase. ERAP1, endoplasmic reticulum aminopeptidase 1. ERAP2, endoplasmic reticulum aminopeptidase 2.

**Table 2.** Cytotoxicity of MIPS2673 against the human HEK293 cell line.

| | % inhibition* | |
|---|---|---|
| Compound | 40 μM | 120 μM |
| MIPS2673 | NI | 89 |
| Puromycin[†] | 100 | 103 |
| Chloroquine | 67 | 65 |
| DHA | 29 | 32 |

*Values represent the mean from two experiments.
[†]Puromycin was tested at 10 μM. NI, no inhibition.

This suggests that the reason for the selectivity between the two enzymes results from the ability (or inability) to access the interior cavity of the *Pf*A-M17 hexamer that houses the active sites.

To investigate the potential for off-target effects, we examined the selectivity of MIPS2673 over several human M1 homologs, including leukotriene A-4 hydrolase (LTA4H), endoplasmic reticulum aminopeptidase 1 (ERAP1), and endoplasmic reticulum aminopeptidase 2 (ERAP2). MIPS2673 showed minimal inhibition of ERAP1 and ERAP2 at concentrations below 500 μM (*Table 1*), whereas MIPS2673 caused >50% inhibition of LTA4H at concentrations above 10 μM (*Table 1*). We also tested MIPS2673 against the HEK293 cell line to further predict potential human cytotoxicity. No cytotoxicity to HEK293 cells was observed up to a concentration of 40 μM. As a comparison, the current antimalarials chloroquine and dihydroartemisinin inhibit HEK293 proliferation by 67% and 29%, respectively, at 40 μM. At 120 μM, MIPS2673 inhibited cellular proliferation by 89% compared to the untreated control (*Table 2*).

## In vitro antiparasitic activity of MIPS2673

Having synthesised a *Plasmodium*-specific M1 aminopeptidase inhibitor, we next explored the activity of MIPS2673 against in vitro blood-stage cultures of *P. falciparum*. We found that MIPS2673 has a 72 hr $EC_{50}$ of 324 nM (250–470 CI) against the laboratory reference strain, 3D7 (*Figure 1C*), compared with the expected low nanomolar $EC_{50}$ achieved with the clinically used artemisinin derivative, artesunate, which was analysed in parallel (*Figure 1—figure supplement 1*). MIPS2673 demonstrated no shift in $EC_{50}$ when tested against a panel of drug-resistant *P. falciparum* lines (*Table 3*). Stage specificity

**Table 3.** MIPS2673 effectiveness against a panel of drug-resistant *P. falciparum* strains*.

| *P. falciparum* strain | Sensitivity profile | $EC_{50}$ (nM)[†] | Fold shift in $EC_{50}$ relative to | |
|---|---|---|---|---|
| | | | NF54 | Dd2 |
| NF54 | Sensitive | 237 | 1.0 | |
| K1 | Decreased sensitivity to chloroquine | 278 | 1.2 | |
| 7G8 | Decreased sensitivity to chloroquine | 293 | 1.2 | |
| TM90C2B | Decreased sensitivity to atovaquone | 194 | 0.8 | |
| Cam3.1 | Decreased sensitivity to artemisinin | 244 | 1.0 | |
| Dd2 | Decreased sensitivity to chloroquine | 220 | 0.9 | 1.0 |
| Dd2 DDD107498 | Decreased sensitivity to eEF2 inhibitor, DDD107498 | 211 | | 1.0 |
| Dd2 MMV390048 | Decreased sensitivity to PI4K inhibitor, MMV390048 | 164 | | 0.7 |
| Dd2 DSM265 | Decreased sensitivity to DHODH inhibitor, DSM265 | 183 | | 0.8 |
| Dd2 GNF156 | Decreased sensitivity to GNF156 | 195 | | 0.9 |
| Dd2 ELQ300 | Decreased sensitivity to cyt bc1 inhibitor, ELQ300 | 174 | | 0.8 |

*72 hr [³H]-hypoxanthine incorporation assay.
[†]Mean values from two biological replicates.

**Table 4.** MIPS2673 effectiveness against *P. falciparum* early (I-III), late (IV-V), and mature (V) gametocyte stages of NF54-pfs16-GFP parasites.

| Compound | $EC_{50}$ (nM)* | | |
| --- | --- | --- | --- |
| | **Early-stage gametocytes** | **Late-stage gametocytes** | **Mature-stage gametocytes** |
| MIPS2673 | 1660±71 | 5300±1923 | 98% (120 μM)[†] |
| Artesunate | 16±5.0 | 26±2.0 | 51±23 |

*$EC_{50}$ values were determined in technical duplicate from two to three biological repeats.
[†]% inhibition at 120 μM ($EC_{50}$ not determined).

assays were performed to identify which stage of asexual development MIPS2673 is most active against. Synchronised ring or trophozoite cultures were exposed to MIPS2673 for 24 hr or 48 hr at 10× $EC_{50}$, then stringently washed to remove the compound. Parasite growth was assessed after 48 hr compared to a vehicle-treated control. This confirmed MIPS2673 is most potent against trophozoite-stage parasites (*Figure 1D*), corresponding to the period of peak *Pf*A-M1 expression (*Allary et al., 2002*) and activity (*Gavigan et al., 2001*). As expected, no stage-dependent difference in activity was observed in parasites exposed to MIPS2673 for 48 hr, as this duration incorporates the full asexual erythrocytic cycle. The control compound, artesunate, was equipotent across all conditions, consistent with the known activity and kinetic profiles for artemisinins (*Yang et al., 2016*).

To visually characterise the effects of MIPS2673 on parasites, we treated synchronised 3D7 cultures at the early ring stage and monitored their development by light microscopy evaluation of Giemsa-stained thin blood smears (*Figure 1E*). We found that MIPS2673 caused significant growth retardation when compared to the untreated control and parasite growth stalled at the ring to trophozoite transition (~16–22 hr post invasion). In contrast, untreated parasites progressed into schizonts and eventually re-invaded red blood cells to commence the next asexual cycle.

Having confirmed activity against asexual stages of *P. falciparum*, we next determined whether MIPS2673 kills the transmissible sexual forms of the parasite, known as gametocytes. We found that MIPS2673 has transmission-blocking activity and is most potent against early gametocytes (stages I-III), compared to late (stages IV-V) and mature (stage V) stages (*Table 4*), but is less potent than the artemisinin derivative, artesunate.

## Identification of molecular targets for MIPS2673 in *P. falciparum* by thermal stability proteomics

Our biochemical studies showing binding and inhibition of recombinant *Plasmodium* metalloamino-peptidases do not rule out the possibility that the antiparasitic activity of MIPS2673 is due to non-specific binding to other parasite metalloproteins or off-target proteins. Genetic target validation was not possible as previous attempts to mutate *Pf*A-M1 were not successful (*Dalal and Klemba, 2007*; *Zhang et al., 2018*), and MIPS2673-resistant parasites were not available. Therefore, we developed orthogonal mass spectrometry-based chemoproteomics approaches to validate that MIPS2673 is on-target and M1-selective in the complex parasite environment (*Figure 2*). We initially developed a streamlined thermal stability proteomics workflow that combined traditional thermal proteome profiling methods with an optimised DIA-liquid chromatography coupled to mass spectrometry (LC-MS/MS) approach (*Siddiqui et al., 2022a*; *George et al., 2023*).

To identify the binding target/s of MIPS2673 in *P. falciparum* asexual blood stages in a proteome-wide manner, we used native parasite lysates, as direct drug-protein interactions are more selectively identified in cellular lysates, rather than live cells that are susceptible to downstream effects of drug action. The parasite lysates were exposed to 1 μM or 4 μM of MIPS2673 or vehicle (DMSO control) for 3 min prior to heating at 60°C, a temperature that should allow detection of most drug-induced protein stabilisation events in an untargeted manner with wide proteome coverage (*Dziekan et al., 2019*; *Dziekan et al., 2020*). After the thermal challenge, the soluble (non-denatured) protein fraction was isolated by ultracentrifugation, digested overnight with trypsin and the peptide mixture analysed directly using global DIA-LC-MS/MS. Proteins detected with significantly higher abundance in treated relative to control samples reflect thermal stabilisation of the target due to ligand binding. To minimise identification of false positive hits, proteins reproducibly stabilised (p<0.05 and

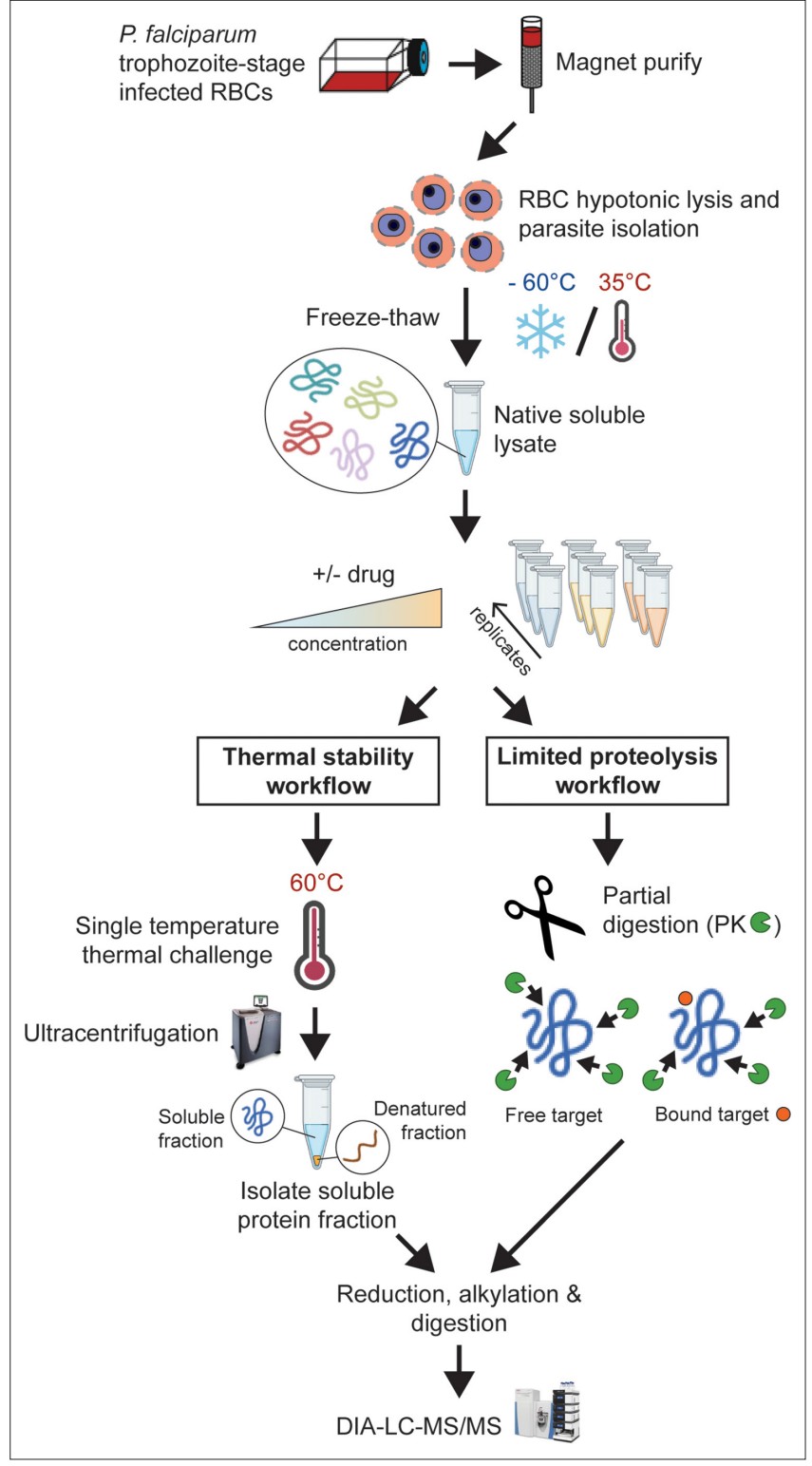

**Figure 2.** Experimental thermal stability and limited proteolysis workflows for unbiased drug target identification in *P. falciparum* lysates. Synchronised trophozoite-stage parasites (30–38 hr post invasion) are isolated from the host red blood cell by hypotonic lysis. Proteins are extracted from cells under native conditions by multiple freeze-thaw cycles. The lysate, comprising the soluble protein fraction, is pooled and distributed to achieve multiple independent incubations per reaction condition for use in either thermal stability or limited proteolysis workflows. For thermal stability proteomics studies, the parasite lysate is subjected to treatment with vehicle (untreated

*Figure 2 continued on next page*

*Figure 2 continued*

control) or drug at two concentrations, followed by thermal challenge at 60°C. The soluble (non-denatured) protein fraction is isolated by ultracentrifugation and digested overnight with trypsin. The resulting peptide mixture is analysed by data-independent acquisition (DIA)-liquid chromatography coupled to mass spectrometry (LC-MS/MS) against an in-house-generated spectral library to determine drug-induced thermal stabilisation. For limited proteolysis studies, ligand-protein interactions are monitored at multiple drug concentrations following double protease digestion. An initial pulse proteolysis with the broad specificity protease, proteinase K (PK), captures local ligand-induced structural alterations of proteins that become differentially susceptible to protease cleavage upon drug binding. Samples then undergo complete digestion under reducing conditions with trypsin/LysC. Differential proteolytic patterns produced upon ligand binding are detected by DIA-LC-MS/MS and analysed against a limited proteolysis spectral library.

---

fold-change ≥1.2 compared to the untreated control) across multiple drug concentrations and experiments were considered drug-interacting proteins. Among 1632 proteins reproducibly detected with a minimum of two peptides across two independent experiments (each with at least three independent incubations of MIPS2673 or vehicle with protein lysate), five proteins were consistently stabilised at both drug concentrations compared to DMSO (*Figure 3A*, *Figure 3—source data 1*, and *Figure 3—source data 2*). Of these, *Pf*A-M1 (PF3D7_1311800) was one of the most significantly stabilised proteins (p<0.01 at 1 μM). *Pf*A-M1 exhibited concentration-dependent stabilisation in the presence of MIPS2673, with an average stabilisation of 1.8-fold and 2.8-fold at 1 μM and 4 μM, respectively, relative to the untreated control (*Figure 3B*). The four other proteins consistently stabilised were not metalloproteins and included two conserved *Plasmodium* proteins with unknown functions (PF3D7_1026000 and PF3D7_0604300), a putative AP2 domain transcription factor (PF3D7_1239200) and a human protein, Ras-related Rab-39A (Q14964) (*Figure 3B*). These hits could represent other MIPS2673-interacting proteins. The *Pf*A-M17 protein was not stabilised by MIPS2673 at either drug concentration after a 60°C thermal challenge (*Figure 3—figure supplement 1*). We previously showed *Pf*A-M17 is stabilised at this temperature in parasite lysates treated with a selective *Pf*A-M17 inhibitor (*Edgar et al., 2022*). Overall, our unbiased thermal stabilisation proteomics approach confirmed MIPS2673 selectively targets the M1 aminopeptidase over M17 and does not bind indiscriminately to parasite metalloproteins.

## Identification of MIPS2673 targets in *P. falciparum* by limited proteolysis coupled mass spectrometry

Having identified several MIPS2673 binding proteins using thermal stability proteomics, we wanted to further investigate the target profile of MIPS2673 with a complementary target deconvolution method. To this end, we developed an efficient protocol for limited proteolysis-based studies of *P. falciparum* and applied it for the unbiased identification of targets of MIPS2673 (*Figure 2*). Native *P. falciparum* lysates were treated for 10 min with different concentrations of MIPS2673 (1 μM or 10 μM in experiment one and 0.1 μM, 1 μM, or 10 μM in experiment two) or vehicle in at least four independent incubations per experiment. Proteome extracts were then subjected to double protease digestion. An initial limited proteolysis with proteinase K for 4 min captured local structural alterations of proteins that become differentially susceptible to protease cleavage upon drug binding. Secondly, samples undergo complete digestion under reducing conditions overnight with trypsin/LysC to make peptides amenable for global proteomics analysis. Differentially abundant proteolytic peptides between MIPS2673 and vehicle-treated samples were then identified on a global scale with DIA-LC-MS/MS. The proteolytic peptide patterns of drug targets should be altered in the treated samples as compound binding prevents protein cleavage by proteinase K, resulting in decreased abundance of peptides with non-tryptic ends or an increase in concentration of the associated fully tryptic peptide (*Figure 4A*). We quantified 26,611 peptides from 2153 proteins in experiment one and 16,662 peptides from 1989 proteins in experiment two (*Figure 4—source data 1*). Each dataset was initially analysed for differentially abundant peptides between each MIPS2673 concentration and vehicle by filtering based upon relative peptide abundance (absolute fold-change >1.5), statistical significance (q<0.01) and proteolytic peptide pattern (i.e. increased fully tryptic or decreased half tryptic). In experiment one and experiment two respectively, approximately 3% of peptides from 379 proteins and 0.6% of peptides from 69 proteins met these thresholds at each drug concentration

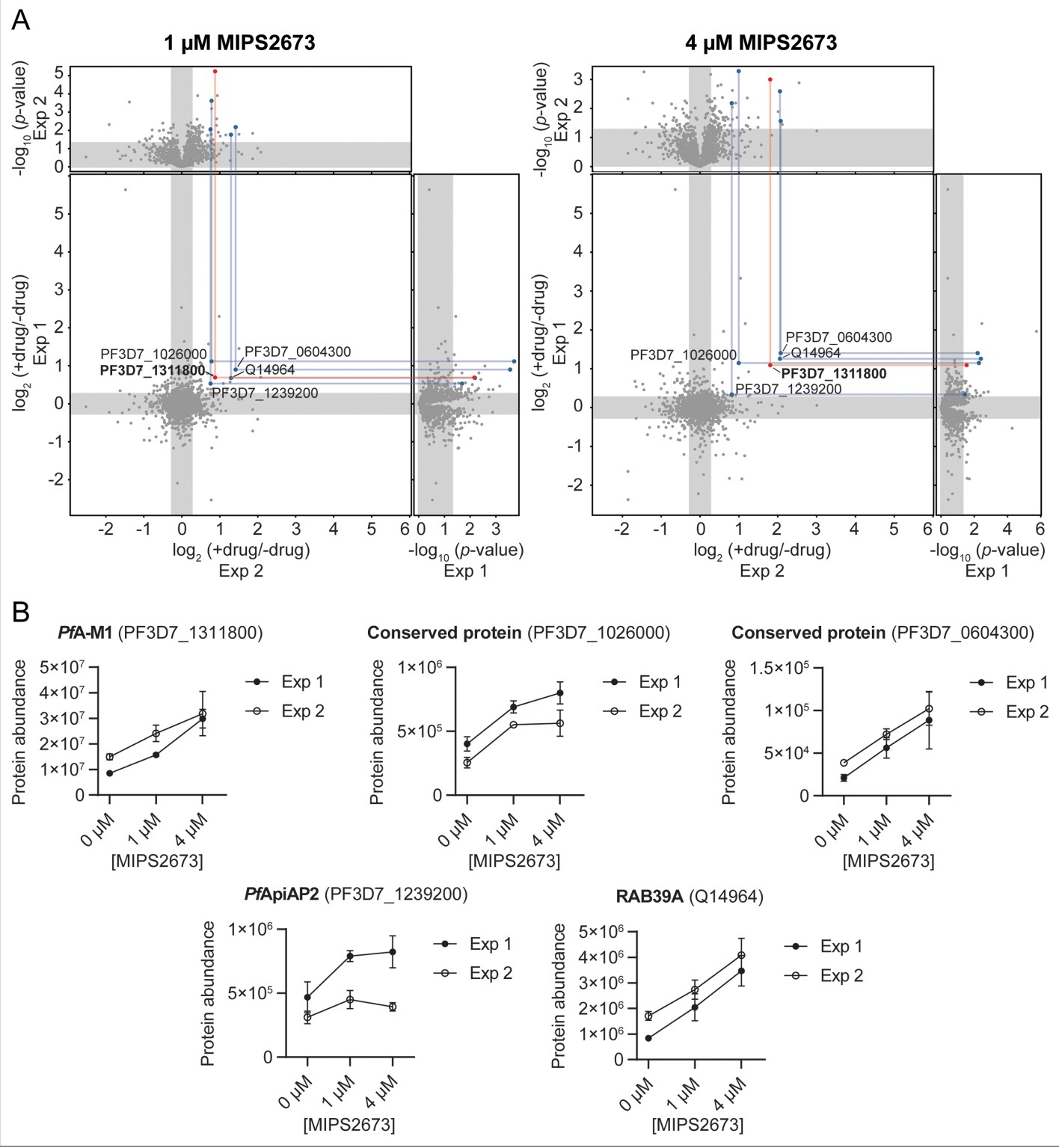

**Figure 3.** Thermal stabilisation of proteins by MIPS2673. (**A**) Paired volcano plots of all proteins detected after a 60°C thermal challenge across two experiments from *P. falciparum* lysates (at least n=3 of independent lysate incubations per condition and experiment) treated with 1 μM (left) and 4 μM (right) MIPS2673. Outside panels show the fold-change in protein abundance for treated relative to untreated samples as a function of statistical significance (p-value) for independent experiments. Each protein is represented with a single data point. Significance cut-offs for thermal stabilisation were p<0.05 (Welch's t-test) and fold-change >1.2 (unshaded regions). The central panel plots protein fold-change from experiments one and two, with proteins in the upper right quadrant reproducibly stabilised by >1.2-fold. Proteins passing both fold-change and statistical cut-offs at both drug

*Figure 3 continued on next page*

*Figure 3 continued*

concentrations and in multiple independent experiments are highlighted. M1-family alanyl aminopeptidase (PF3D7_1311800) is in red and all other proteins passing cut-offs are in blue. (**B**) Individual profiles for the five proteins reproducibly stabilised by MIPS2673 after a 60°C thermal challenge. Two independent experiments are shown. Data are mean protein abundances in the absence and presence of MIPS2673 (± SD) from four replicates.

The online version of this article includes the following source data and figure supplement(s) for figure 3:

**Source data 1.** Thermal stability proteomics datasets related to *Figure 3*.

**Source data 2.** Thermal stability proteomics joint volcano plot data related to *Figure 3*.

**Figure supplement 1.** Individual thermal stabilisation profile of *Pf*A-M17 leucyl aminopeptidase after *P. falciparum* lysate treatment with MIPS2673.

and were considered significant (*Figure 4B*). After prioritising targets based upon the number of significant peptides detected per protein (Fisher's exact test, Bonferroni corrected p<0.05), we identified seven putative drug targets in experiment one and only one putative target in experiment two (*Figure 4C* and *Figure 4—source data 2*). Interestingly, among the seven possible target proteins identified by limited proteolysis in experiment one, five were metalloproteins. These included *Pf*A-M1, three other aminopeptidases – M17 leucyl aminopeptidase, aminopeptidase P (*Pf*APP), and M18 aspartyl aminopeptidase (*Pf*A-M18) – and adenosine deaminase (*Pf*ADA) (*Figure 4C*). The expected target, *Pf*A-M1 aminopeptidase, was the only protein reproducibly identified as the MIPS2673 target across both independent limited proteolysis experiments. Indeed, the consistent identification of *Pf*A-M1 as the only significant protein in both limited proteolysis coupled with mass spectrometry (LiP-MS) and thermal stability proteomics methods strongly supports *Pf*A-M1 being the primary target of MIPS2673.

## Features of structurally significant *Pf*A-M1 LiP-MS peptides

The identification of drug targets with LiP-MS typically identifies structurally perturbed peptides located in very close proximity to the ligand binding site (*Piazza et al., 2020*; *Piazza et al., 2018*) and we hypothesised that this would also be the case in our study. Among the 108 *Pf*A-M1 peptides reproducibly detected across both LiP-MS experiments, we identified nine structurally significant LiP-MS peptides (q<0.01 and absolute fold-change >1.5 at all drug concentrations in both experiments) (*Figure 5A*) and mapped these to our *Pf*A-M1 crystal structure with MIPS2673 bound (PDB ID: 8SLO). We measured the minimum distance between atoms of the significant peptides and those of MIPS2673 and found that structurally significant peptides identified with LiP-MS were frequently located in very close proximity to MIPS2673. The median minimum distance between the significant LiP-MS peptides and bound MIPS2673 was 6.5 Å, significantly less than the 13.2 Å median distance for all detected *Pf*A-M1 peptides (p=0.025) (*Figure 5—figure supplement 1*). Among the nine structurally significant LiP-MS peptides, four contained atoms within Van der Waals distance (<4 Å) of MIPS2673. In contrast, the median minimum distance of structurally perturbed LiP peptides of other putative MIPS2673-interacting proteins identified with either thermal stability proteomics or LiP-MS (experiment one) were not significantly closer to expected binding sites when compared to all peptides detected, except for significant LiP peptides from *Pf*A-M17 (median minimum distance of 8.56 Å vs 17.47 Å, p=0.045) (*Figure 5—figure supplement 2*). Based on these observations, we hypothesised that significantly dysregulated LiP-MS peptides could estimate the known MIPS2673 binding site on *Pf*A-M1. A distance of 6.44 Å from MIPS2673 was used to define the binding cleft boundary (*Piazza et al., 2018*) and included the *Pf*A-M1 residues known to interact with bound MIPS2673. Six of the nine significant LiP peptides overlapped with, or were within 4 Å of this binding cleft. The median distance between the atoms of significant LiP peptides and of cleft residues was 2.6 Å, compared to a median distance of 8.2 Å for all *Pf*A-M1 peptides (p=0.029) (*Figure 5B*). Of the three significant LiP peptides located >4 Å from the operational binding site, two are located at the C-terminal opening in domain IV that forms a channel leading towards the active site (*McGowan et al., 2009*). To understand if this type of data could approximate a ligand binding site correctly, we calculated the centre of mass of the atoms of the nine structurally significant LiP peptides and represented this as a geometric point within the *Pf*A-M1 structure and then compared this to the known binding site (*Figure 5C*). The minimum distance between this point and MIPS2673 was 5.2 Å and the MIPS2673 centre of mass neighbourhood (residues within 6.44 Å of the centre of mass) overlapped with the binding site

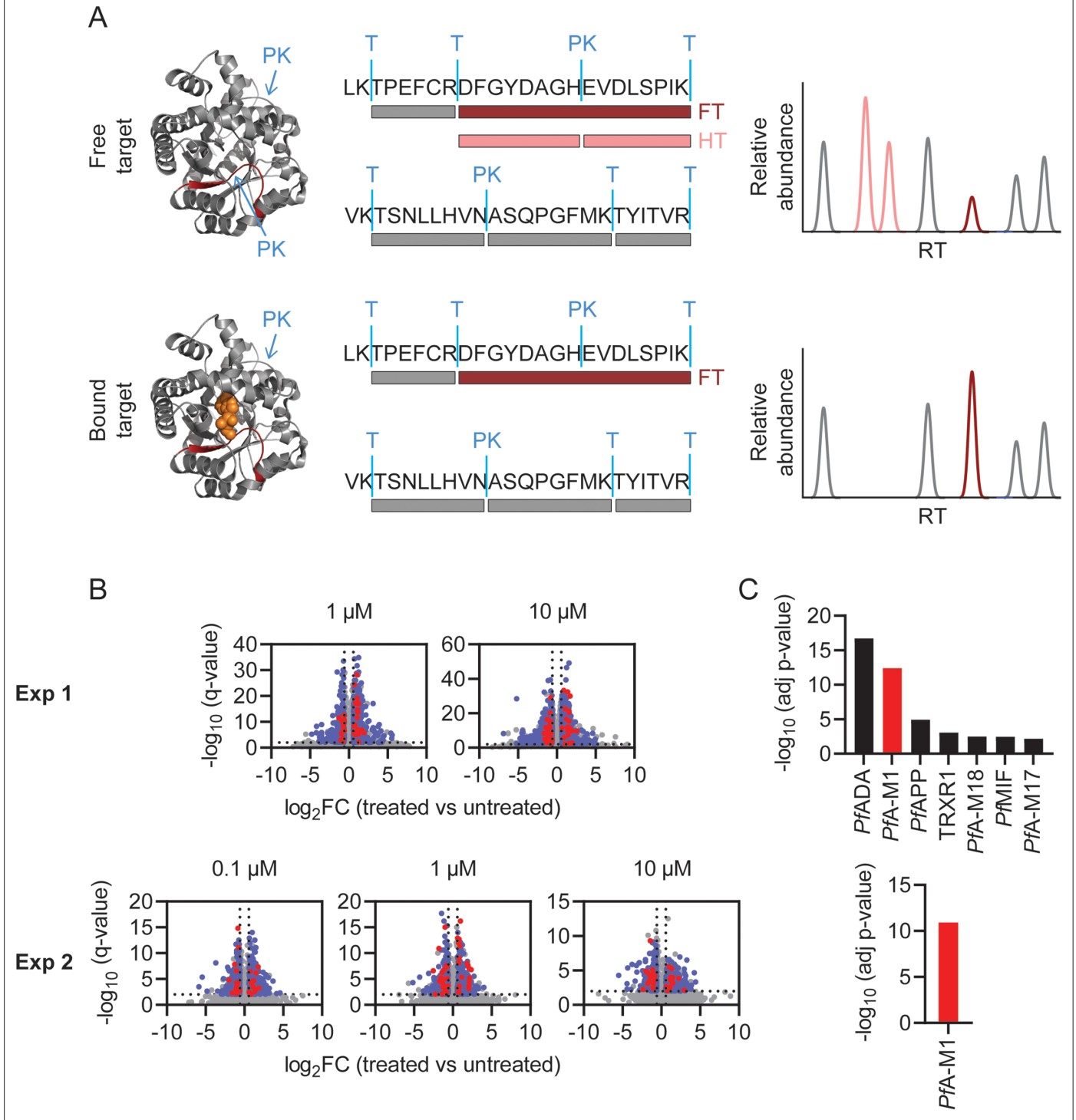

**Figure 4.** Identification of MIPS2673 target proteins using limited proteolysis coupled with mass spectrometry (LiP-MS). (**A**) Principle of LiP-MS analysis. Ligand binding to a protein alters the local proteolytic susceptibility and prevents protein cleavage by proteinase K. In the bound state this results in decreased abundance of peptides with non-tryptic ends (half tryptic, HT) shown in pink and/or an increase in concentration of the corresponding fully tryptic (FT) peptide, shown in red. Peptides unaffected by ligand binding (grey) are unchanged in abundance. (**B**) Volcano plots of two independent LiP-MS experiments (at least n=3 independent lysate incubations per condition and experiment). Each data point represents a detected peptide. Fold-changes (FC) in peptide abundance in treated vs untreated samples are shown as a function of significance (q<0.01) at the MIPS2673 concentrations shown. Peptides highlighted in blue represent LiP peptides that were significant across all concentrations tested within an experiment. Red peptides represent significant LiP peptides from *Pf*A-M1. (**C**) Short-list of putative MIPS2673 targets identified in each LiP-MS experiment after prioritising targets

*Figure 4 continued on next page*

*Figure 4 continued*

based upon the number of significant peptides detected per protein. Significance was determined using a Fisher's exact test and Bonferroni corrected p<0.05. The expected target *Pf*A-M1 aminopeptidase is shown in red.

The online version of this article includes the following source data and figure supplement(s) for figure 4:

**Source data 1.** Limited proteolysis coupled with mass spectrometry (LiP-MS) proteomics datasets related to *Figure 4* and *Figure 5*.

**Source data 2.** Limited proteolysis coupled with mass spectrometry (LiP-MS) Fisher's exact test p-values related to *Figure 4*.

**Source data 3.** Experiment one: limited proteolysis coupled with mass spectrometry (LiP-MS) gene ontology (GO) molecular function enrichment analysis related to *Figure 4*.

**Figure supplement 1.** Molecular function gene ontology (GO) enrichment analysis of the MIPS2673 experiment one: limited proteolysis coupled with mass spectrometry (LiP-MS) dataset.

(*Figure 5C*). This indicated that our LiP-MS approach provides a good approximation of the MIPS2673 binding site with its target, *Pf*A-M1.

## Untargeted metabolomics analysis of *P. falciparum* infected red blood cells treated with MIPS2673

To further determine the specificity of MIPS2673 for *Pf*A-M1 and whether it induces off-target effects, we performed untargeted metabolomics on infected red blood cells treated with 1 μM of MIPS2673 (3× EC$_{50}$ value) for 1 hr and compared the profile to vehicle (DMSO) control (four to nine biological replicates). Heatmap analysis of relative abundances of all putative metabolites revealed that treatment with MIPS2673 disproportionally impacted peptide metabolism (*Figure 6—figure supplement 1*). Of the 201 putative peptides identified, 97 were significantly dysregulated (p<0.05). The majority of dysregulated peptides were significantly increased (fold-change >1.5 and p<0.05) in abundance in MIPS2673-treated cultures compared to DMSO control, indicative of aminopeptidase inhibition. Targeted analysis of the 97 increased peptides in drug-treated cultures revealed that the majority were likely derived from Hb chains (α and β) (*Figure 6A*). Mapping to Hb sequences was possible for 24 peptides using MS/MS spectra to confirm peptide sequences (*Figure 6B*; green dots). For the remaining putative peptides identified by accurate mass, but for which MS/MS spectra could not be obtained, we assessed whether any peptide isomeric to the putative peptide could be mapped to Hb (*Figure 6B*; orange dots for peptides mapping to Hb chains and blue dots for peptides that could not be mapped). Overall, ~80% of significantly dysregulated peptides could be mapped to one of the Hb chains, with nearly all increasing in abundance following treatment with MIPS2673 compared to DMSO control. To further validate that the majority of elevated peptides are likely to be Hb-derived, we repeated this same analysis for each of the ~4700 proteins identified in our recent comprehensive proteomic analysis of *P. falciparum*-infected red blood cells (*Siddiqui et al., 2022a*). We quantified the number of peptide matches to each protein and then divided by protein length to yield a normalised estimate of the similarity of each protein to our significantly dysregulated peptides. By this measure, Hb chains α and β were the most highly matched protein compared to the remaining infected red blood cell proteome (*Figure 6C*; red bars), indicating that MIPS2673 predominantly, but not exclusively, disrupts Hb digestion.

We have recently reported that an inhibitor of *Pf*A-M17 (MIPS2571) also predominantly disrupted metabolism of short Hb-derived peptides, which was a metabolic signature consistent with genetic knockdown of *Pf*A-M17 (*Edgar et al., 2022*). However, a direct comparison of peptide perturbations induced by MIPS2673 and MIPS2571 (data from *Edgar et al., 2022*) showed clearly distinct peptide profiles. Whilst many peptides accumulated with both inhibitors, the extent of peptide accumulation differed, and a subset of short basic peptides (containing Lys or Arg) were elevated following exposure to MIPS2673, but not the *Pf*A-M17 inhibitor (*Figure 6A*). Furthermore, the peptide changes observed following treatment with MIPS2673 differ substantially from the peptide changes observed following treatment of *P. falciparum*-infected red blood cells with other potent antimalarials such as artemisinins (*Creek et al., 2016*; *Giannangelo et al., 2020*) and mefloquine (*Birrell et al., 2020*). Those antimalarials induce significant depletion of Hb-derived peptides, in contrast to the accumulation observed with the aminopeptidase inhibitors. Overall, the metabolic profile induced by MIPS2673 is unique, and consistent with specific inhibition of *Pf*A-M1.

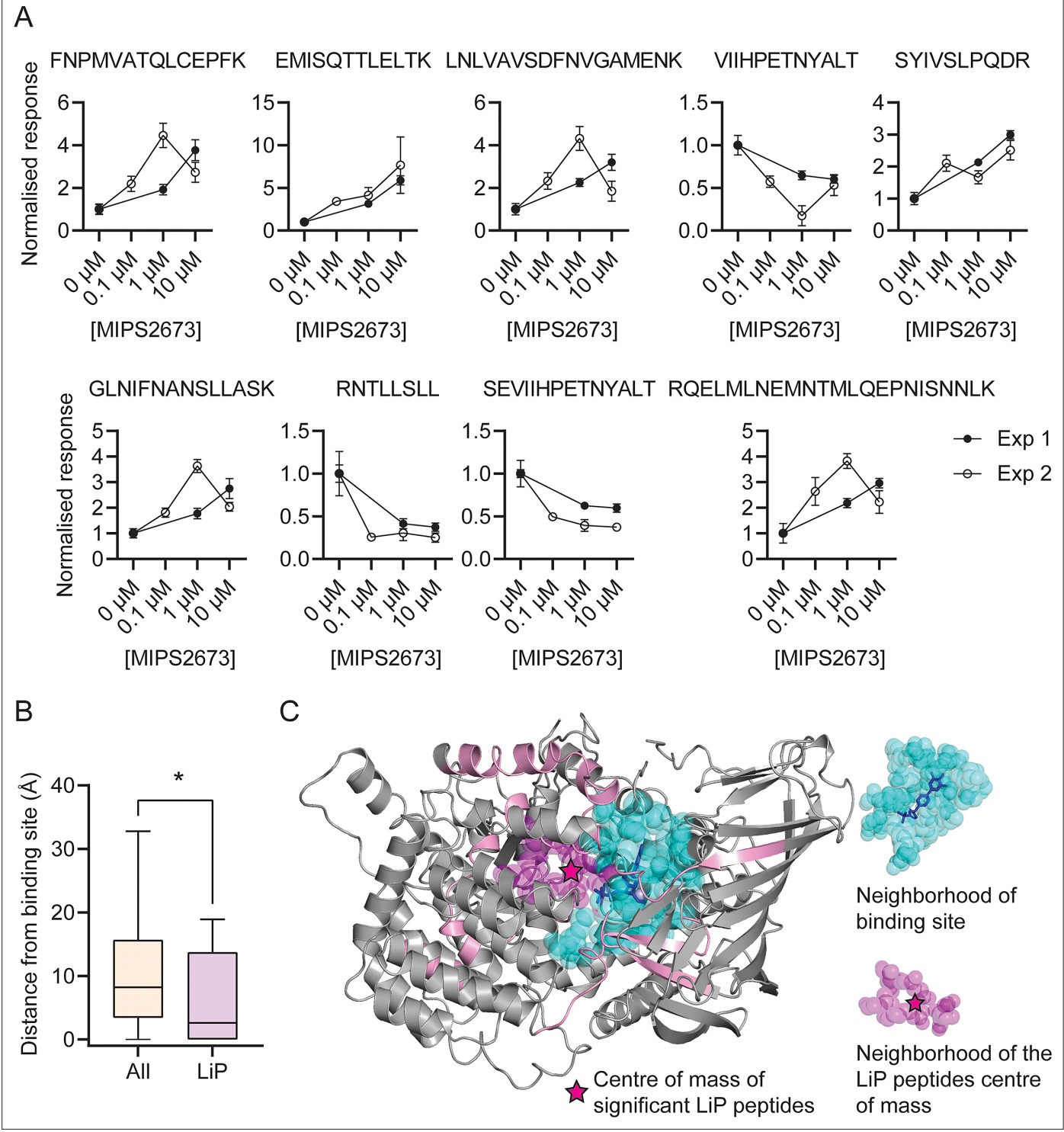

**Figure 5.** Features of structurally significant *Pf*A-M1 limited proteolysis coupled with mass spectrometry (LiP-MS) peptides. (**A**) Relative abundance of the significant LiP peptides commonly identified across two LiP-MS experiments following *P. falciparum* proteome lysate treatment with MIPS2673. The mean ± SEM of at least three independent lysate incubations per condition and experiment are shown. (**B**) Median distances between atoms of the nine significant LiP peptides or all detected *Pf*A-M1 peptides and the *Pf*A-M1 binding cleft residues. *p<0.05, Mann-Whitney test. (**C**) MIPS2673 binding site on *Pf*A-M1 determined by X-ray crystallography (PDB: 8SLO), and approximation of the MIPS2673 binding site using the significant *Pf*A-M1 LiP peptides and centre of mass calculation. The significant LiP peptides are mapped onto the *Pf*A-M1 structure with MIPS2673 bound and are shown in pink. The drug ligand is shown in blue and the centre of mass of the significant LiP peptides is shown by a magenta star. The area in cyan represents

*Figure 5 continued on next page*

*Figure 5 continued*

the neighbourhood of the drug binding site determined as residues within 6.44 Å of bound MIPS2673. The neighbourhood of the LiP peptide centre of mass (residues within 6.44 Å of the centre of mass) is depicted in magenta.

The online version of this article includes the following figure supplement(s) for figure 5:

**Figure supplement 1.** Minimum distance of significant limited proteolysis coupled with mass spectrometry (LiP-MS) or all other *Pf*A-M1 peptides from bound MIPS2673.

**Figure supplement 2.** Minimum distance measurements and mapping of significant limited proteolysis coupled with mass spectrometry (LiP-MS) peptides to structures of other putative MIPS2673-interacting proteins.

## Discussion

Here, we report the design of MIPS2673, a hydroxamic acid-based inhibitor that targets both the *P. falciparum* (*Pf*A-M1) and *P. vivax* (*Pv*A-M1) M1 aminopeptidases with excellent selectivity over *Plasmodium* M17 and human M1 homologs. We developed orthogonal mass spectrometry-based chemoproteomics methods based on thermal stability and limited proteolysis to validate, in an unbiased manner, that MIPS2673 engages *Pf*A-M1 in a parasite lysate. This was further supported by untargeted metabolomic profiling that revealed significant accumulation of short peptides in MIPS2673-treated parasites, consistent with *Pf*A-M1 inhibition. MIPS2673 is active against a panel of drug-resistant *P. falciparum* strains, including parasites with decreased susceptibility to the current frontline antimalarials, has potential to treat both active disease (asexual blood stage activity) and block parasite transmission (anti-gametocyte activity), and is non-cytotoxic to the human HEK293 cell line. Together, our studies confirm *Plasmodium* M1 alanyl aminopeptidase as the target of MIPS2673 and validate selective inhibition of this enzyme as a promising antimalarial strategy.

The gene encoding *Pf*A-M1 was previously shown to be refractory to knockout and is predicted to be essential for blood-stage growth, making it an attractive antimalarial drug target (**Dalal and Klemba, 2007**; **Zhang et al., 2018**). To confirm that MIPS2673 inhibited *Pf*A-M1 in the parasite, we initially developed a streamlined workflow that combined traditional thermal stability proteomics methods with a DIA-LC-MS/MS approach. Thermal proteome profiling was recently adapted for *P. falciparum* and applied for antimalarial drug target identification using data-dependent acquisition (DDA)-based LC-MS/MS analysis (**Dziekan et al., 2019**; **Dziekan et al., 2020**; **Milne et al., 2022**). Here, protein thermal stabilisation is typically monitored in the presence or absence of compound over a range of temperatures (thermal proteome profiling), or at multiple drug concentrations at a single temperature (isothermal dose response). The extensive samples generated by these approaches require multiplexing with expensive isobaric stable isotope labelling reagents and offline fractionation to minimise sample complexity prior to DDA-LC-MS/MS analysis. Furthermore, DDA-based proteomics is prone to inconsistent peptide detection, which limits the number of proteins that can be reproducibly identified and quantified. In contrast, using DIA analyses in thermal proteome profiling was recently shown to be an effective and economical alternative to traditional DDA-based thermal shift quantitation workflows (**George et al., 2023**). DIA-based protein quantification also results in high run-to-run reproducibility and a comprehensive dataset that is not biased towards highly abundant proteins (**Siddiqui et al., 2022b**; **Gillet et al., 2012**). Our DIA-based thermal stability proteomics workflow identified approximately 47% of the detectable *P. falciparum* blood-stage proteome (**Siddiqui et al., 2022a**), with the undetected proteins primarily comprised of membrane-bound proteins that are not readily extracted with detergent-free buffers required for stability assays of soluble proteins. This proteome coverage is comparable to previously reported DDA thermal proteomics studies of *P. falciparum* (**Dziekan et al., 2019**; **Dziekan et al., 2020**; **Milne et al., 2022**), but our method avoids the need for testing over a wide range of concentrations or temperatures, isobaric tags, and offline fractionation steps that increase the time, cost, and complexity associated with previously published methods. Since in bacterial and other eukaryotic cellular systems, LiP-MS has proven useful for revealing direct drug targets, protein interactions with endogenous metabolites, and ligand binding sites at peptide-level resolution (**Piazza et al., 2020**; **Piazza et al., 2018**), we also explored LiP-MS based on DIA-LC-MS/MS as a complementary approach to demonstrate MIPS2673 target engagement in malaria parasites.

Both thermal stability proteomics and LiP-MS strategies support unbiased deconvolution of drug mechanisms on a system-wide scale. Whilst our in vitro enzyme assay data demonstrated selective inhibition of recombinant *Pf*A-M1 over *Pf*A-M17 (fourfold), with a $K_i$ for *Pf*A-M1 inhibition similar to

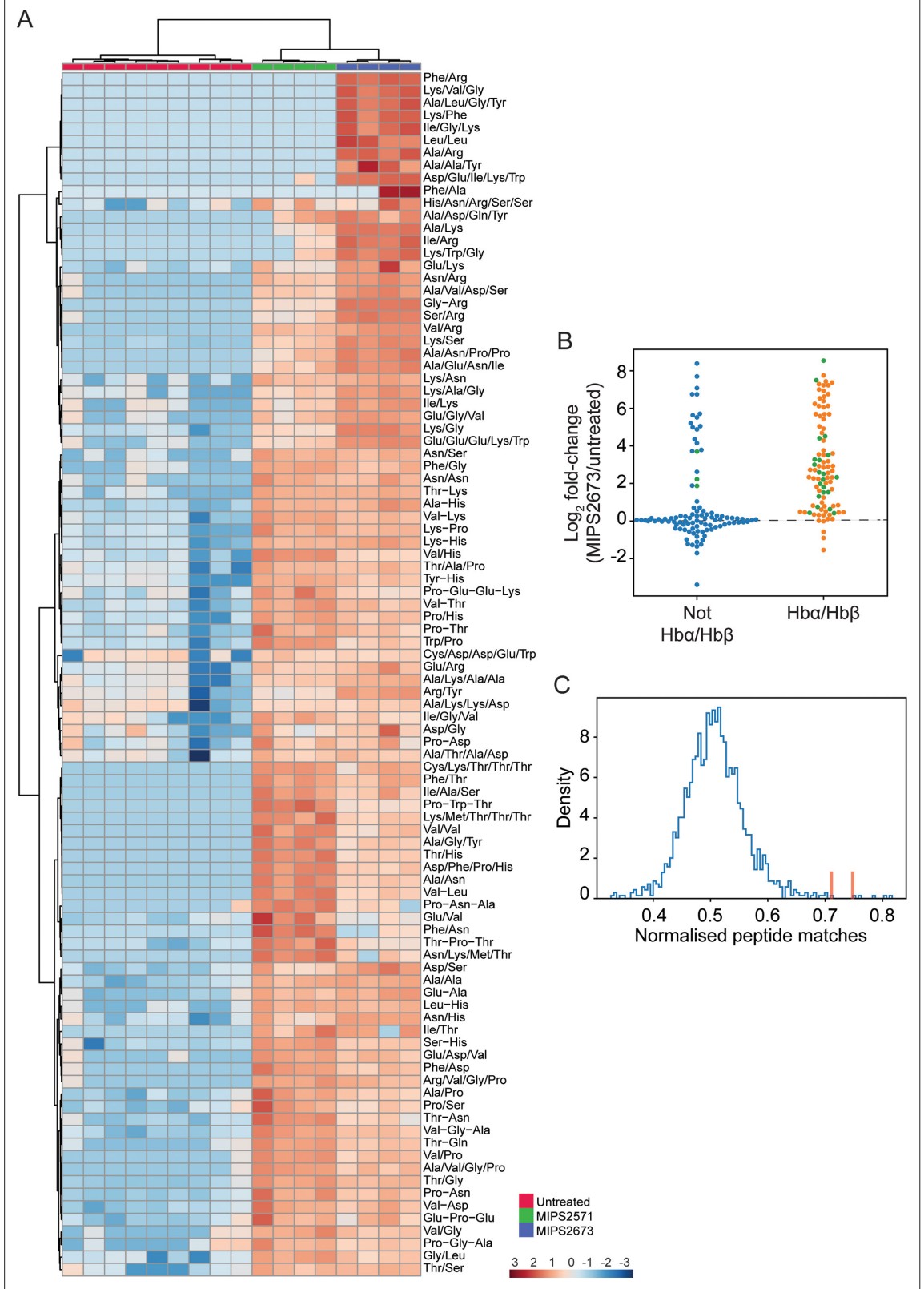

**Figure 6.** Targeted analysis of significantly dysregulated peptides (p<0.05) following treatment with 1 μM of MIPS2673 (*Pf*A-M1 inhibitor) for 1 hr compared to MIPS2571 (*Pf*A-M17 inhibitor) and DMSO control. (**A**) Hierarchical clustering of the 97 peptides significantly dysregulated after treatment with MIPS2673 (fold-change >1.5 and p<0.05); four biological replicates for MIPS2673 and MIPS2571 (data from *Edgar et al., 2022*) and nine biological replicates for DMSO control. Vertical clustering displays similarities between samples, while horizontal clusters reveal the relative abundances (median

*Figure 6 continued on next page*

*Figure 6 continued*

normalised) of the 97 peptides. The color scale bar represents log2 (mean-centred and divided by the standard deviation of each variable) intensity values. Peptides with hyphen (-) notation indicate confirmed sequence by MS/MS. Peptides with slash (/) notation indicate putative amino acid composition (accurate mass), without confirmed sequence order. (**B**) Differential enrichment of all (201) putatively identified peptides that could (orange dots) or could not (blue dots) be derived from haemoglobin (Hb) α and β. Green dots are peptides that have MS/MS spectra and their sequences have been confirmed. (**C**) Histogram of the sequence similarity of ~4700 proteins present in *P. falciparum* infected red blood cells to the peptides significantly dysregulated by treatment with MIPS2673. Here, sequence similarity is quantified as the number of times a significantly perturbed peptide matches a given protein, normalised by protein length. The Hb chains α and β are highlighted in red bars.

The online version of this article includes the following figure supplement(s) for figure 6:

**Figure supplement 1.** Untargeted metabolomics analysis of 3D7 parasites treated with 1 µM of MIPS2673.

---

the $EC_{50}$ for killing *P. falciparum* in vitro, this does not rule out inhibition of *Pf*A-M17 or other protein targets within parasites, especially when the $EC_{50}$ concentration approaches the *K*i for *Pf*A-M17 inhibition. Thermal stability proteomics and limited proteolysis coupled with global DIA-LC-MS/MS analysis reproducibly identified *Pf*A-M1 as the target of MIPS2673 from approximately 2000 detected proteins. This validated our chemoproteomics strategy for detection of ligand-protein interactions in complex proteomes and confirmed the on-target activity and M1 selectivity of MIPS2673. Even at 3× $EC_{50}$, and concentrations expected to inhibit recombinant *Pf*A-M17, there were no off-targets consistently identified across both thermal stability proteomics and LiP-MS methods, highlighting the importance of using complementary approaches to map drug interactions and discriminate true protein interactors from false positive identifications.

Our structural analysis showed that MIPS2673 acts by chelating the catalytic zinc ion of the *Pf*A-M1 active site, similar to all other inhibitors produced to date (*Skinner Adams et al., 2007*; *Harbut et al., 2011*; *Deprez-Poulain et al., 2012*; *Drinkwater et al., 2016b*; *Vinh et al., 2019*; *Poreba et al., 2012*; *McGowan et al., 2009*; *Flipo et al., 2007*; *Harbut et al., 2008*; *Velmourougane et al., 2011*). A strategy targeting the catalytic zinc may present challenges for achieving selectivity over host zinc-dependent enzymes, but MIPS2673 demonstrated excellent selectivity over several human M1 homologues, providing rationale for further development of M1 selective inhibitors that present a low risk of host toxicity. Our LiP-MS analysis also supported a zinc-dependent binding mechanism. While other stability-based proteomics methods for drug target identification, such as thermal proteome profiling, map drug-protein interactions by measuring variations in thermal stability of whole proteins, they lack the structural resolution of LiP-MS, which measures local changes in proteolytic susceptibility. Most of the structurally significant *Pf*A-M1 peptides we identified by LiP-MS were within 5 Å of the MIPS2673 binding site determined by X-ray crystallography. The structurally altered peptides identified with LiP-MS included the peptide $S_{292}$EVIIHPETNYALT$_{305}$, which guards the entrance pathway to the active site and is likely involved in substrate recognition and catalytic competence of the enzyme structure (*Papakyriakou and Stratikos, 2017*) as well as the peptide $L_{449}$NLVAVSDFNVGA-MENK$_{465}$, which contains the highly conserved GAMEN exopeptidase motif (*McGowan et al., 2009*). The GAMEN motif functions to bring substrate to the active site metal for catalysis and contributes hydrogen bonds for ligand binding (*McGowan et al., 2009*). Interestingly, we also identified several structurally significant LiP-MS peptides distal (>5 Å) to this location. Ligands and substrates access the *Pf*A-M1 active site via a long C-terminal channel within domain IV and their migration is orchestrated by several transient states which are stabilised by interactions with specific residues of the domain, including $Arg^{969}$, $Arg^{489}$, $Lys^{849}$, $Lys^{907}$, $Glu^{850}$, $Asp^{830}$, $Glu^{572}$, and $Asp^{581}$ (*Moore et al., 2018*). We identified the structurally significant peptides $S_{822}$YIVSLPQDR$_{831}$ and $R_{898}$NTLLSLL$_{905}$ that are located >5 Å from the MIPS2673 binding site at the entrance to the channel cavity. The SYIVSLPQDR and RNTLLSLL peptides contain, or are in very close proximity to, the $ASP^{830}$ and $Lys^{907}$ residues, respectively, which may have a role in regulating recognition and initial passage and migration of ligands into the channel cavity (*Moore et al., 2018*). Together, this demonstrates the potential of LiP-MS not only for unbiased identification of protein targets of drugs, but also for estimating drug binding and interaction sites, which is not possible with many other target deconvolution approaches.

*Pf*A-M1 has been shown to play an essential role in the terminal stages of Hb digestion and the generation of an amino acid pool within the parasite cytosol (*Harbut et al., 2011*). Turnover of Hb into oligopeptides and amino acids peaks during the trophozoite stage when *Pf*A-M1 expression and activity is highest. In agreement with *Pf*A-M1 being the target of MIPS2673, we showed that the

trophozoite stage coincides with the period that parasites are most susceptible to MIPS2673, and that the growth of MIPS2673-treated parasites becomes stalled at the ring to trophozoite transition. Inhibition of *Pf*A-M1 using an activity-based probe based on the bestatin scaffold resulted in trophozoite-stage stalling of parasite growth (*Harbut et al., 2011*). However, the activity-based probe also caused digestive vacuole swelling, a phenotype that we did not observe. Digestive vacuole swelling is commonly associated with inhibition of proteases such as the falcipains that are involved in initiating Hb digestion within the parasite digestive vacuole. Since *Pf*A-M1 is a cytoplasmic protein (*Mathew et al., 2021*), which is presumed to digest short peptides exported from the digestive vacuole, specific inhibition of this enzyme with MIPS2673 is unlikely to cause digestive vacuole swelling. Instead, the phenotype previously observed with the activity-based probe may be the result of off-target effects. Untargeted metabolomics analysis revealed significant accumulation of short peptides following treatment of parasite cultures with MIPS2673. Many of these accumulated peptides likely originate from Hb as MS/MS and accurate mass isomer analysis shows the sequence of these peptides to be more similar to Hb than other host or parasite proteins. A subset of the dysregulated peptides cannot originate from Hb and may instead derive from other host proteins or the turnover of parasite peptides. While we cannot exclude altered processing of parasite-derived peptides contributing to MIPS2673 activity, starving the parasite of key Hb-derived amino acids appears to be the most likely mechanism leading to parasite death after MIPS2673 treatment.

Several aminopeptidases in addition to *Pf*A-M1 are involved in the terminal stages of peptide processing within malaria parasites. These include the essential aminopeptidases, *Pf*A-M17 and aminopeptidase P (*Pf*APP), and the dispensable enzyme, *Pf*A-M18, all of which share a metal-dependent catalytic mechanism. Although no off-targets were consistently identified in our study, four of the seven putative hits identified in our first LiP-MS experiment were metalloaminopeptidases (*Pf*A-M1, *Pf*A-M17, *Pf*A-M18, and *Pf*APP), with only *Pf*A-M1 reproducibly identified as the primary MIPS2673 target across both LiP-MS experiments. *Pf*A-M18 and *Pf*APP are unlikely to be true off-targets that contribute to the antimalarial mechanism in parasites as we found MIPS2673 is non-inhibitory towards recombinant versions of these enzymes, even at concentrations >120-fold above the $EC_{50}$ for in vitro parasite killing. We found that the median minimum distance for significant LiP peptides from *Pf*A-M17 is closer to the active site than the median distance of all detected peptides (8.56 Å vs 17.47 Å, p=0.045). This suggests that MIPS2673 is interacting close to the *Pf*A-M17 active site. However, *Pf*A-M17 was not stabilised by MIPS2673 in our thermal stability proteomics dataset at 60°C, a temperature that we previously showed stabilised *Pf*A-M17 when bound to an inhibitor (*Edgar et al., 2022*), and specific loss of *Pf*A-M17 function results in multiple membrane-bound digestive vacuoles per parasite, a phenotype not observed in parasites treated with MIPS2673. These findings argue against *Pf*A-M17 being a true off-target. Instead, peptide substrates accumulating due to primary inhibition of *Pf*A-M1 may interact with *Pf*A-M17, leading to structural changes around the enzyme active site that are detected by LiP-MS. Furthermore, distinct peptide profiles for MIPS2673 and a *Pf*A-M17 inhibitor were observed in our metabolomics dataset, with MIPS2673 showing specific accumulation of basic peptides, which agrees with the broad substrate specificity of *Pf*A-M1. In our first LiP-MS experiment, we also identified the zinc metalloprotein adenosine deaminase (*Pf*ADA) as a possible MIPS2673 off-target. *Pf*ADA is one of three purine salvage enzymes and catalyses deamination of adenosine to inosine, as well as the conversion of 5'-methylthioadenosine, derived from polyamine biosynthesis, into 5'-methylthioinosine. Investigation of metabolites identified in our metabolomics experiment revealed no significant change in purine or polyamine pathway metabolites, indicating that within 1 hr MIPS2673 has no functional impact on these pathways in parasites. Although it is possible that secondary mechanisms, such as impacts to the purine or polyamine pathway, become evident at longer durations of treatment. Interestingly, reanalysis of this dataset from experiment one with a less stringent statistical cut-off (Fisher's exact test, uncorrected p<0.05) revealed 47 putative targets (*Figure 4—source data 2*) that were significantly enriched for metallopeptidase (p=0.022), aminopeptidase (p=2.1E-4), aspartic-type endopeptidase (p=7.34E-3), or threonine-type endopeptidase (p=1.2E-11) gene ontology (GO) terms (*Figure 4—figure supplement 1*). Proteins within the metal ion binding GO term were not significantly enriched (p=0.985) (*Figure 4—source data 3*), confirming that MIPS2673 does not interact indiscriminately with metalloproteins. Whilst this extended list of putative peptidase-related targets may represent plausible secondary targets – either stabilised by MIPS2673 itself or by peptides accumulating due to primary inhibition of *Pf*A-M1 – they do not appear

responsible for the antiparasitic activity of MIPS2673, as stabilisation of these additional putative targets was not reproducible across independent experiments, and the relevance of the most significant of these (*Pf*A-M17, *Pf*A-M18, *Pf*APP, and *Pf*ADA) is refuted by our thermal stability, metabolomics, and recombinant enzyme inhibition data.

Thermal stability proteomics identified *Pf*A-M1 and four other proteins that appeared to be stabilised by MIPS2673. Of these four additional proteins, none were metalloprotein or peptidase off-targets, and only the putative AP2 domain transcription factor (PF3D7_1239200) is a viable drug target based on its predicted essentiality for blood-stage growth (*Zhang et al., 2018*). This protein is thought to be localised to the parasite nucleus and involved in heterochromatin-associated gene expression (*Shang et al., 2022*). *Pf*A-M1 was previously thought to have nuclear localisation (*Ragheb et al., 2011*) but is now widely accepted to be cytosolic and no function within the parasite nucleus is currently known. With the exception of PF3D7_1239200, it is unlikely that these proteins contribute to MIPS2673 activity in parasites and it is possible that the cell lysis step required to generate a parasite lysate for thermal proteomics (and LiP-MS) introduces artefacts by allowing novel protein interactions or compound access to previously compartmentalised proteins. While the proteins identified individually by thermal stability proteomics and LiP-MS could be MIPS2673-interacting proteins, their lack of consistency across assays suggests that they are likely false discoveries. *Pf*A-M1 was the sole protein reproducibly identified across both unbiased target identification methods, indicating it is the primary target in parasites. However, it should be noted that thermal stability proteomics and LiP-MS rely on complementary stabilisation principles and not all targets will necessarily be amenable to detection by both of these methods. Detecting ligand-induced stabilisation of a target will depend on a number of factors, e.g., the biophysical properties of the target protein, the experimental design, and the depth and coverage of the proteome. Combining multiple complementary approaches will maximise the likelihood of uncovering the complete ligand-protein interaction landscape.

Collectively, we have demonstrated selective targeting of *Plasmodium* M1 alanyl aminopeptidase as an effective antimalarial strategy and validated MIPS2673 as a *Pf*A-M1 inhibitor through unbiased MS-based chemoproteomics. This work provides a basis for continued development of inhibitors against *Plasmodium* M1 alanyl aminopeptidase in an era when new therapeutics with novel mechanisms of action are urgently needed to combat widespread emergence of malaria parasite drug resistance. More broadly, our study demonstrates the power of mass spectrometry-based techniques to simultaneously probe whole proteomes in an unbiased manner and identify direct drug-protein interactions. The unbiased and orthogonal MS-based proteomics methods described here support investigation of drug mechanisms on a system-wide scale without the need for a specific prior hypothesis about the function of a compound. This makes LiP-MS and thermal stability proteomics ideally suited to reveal the molecular targets of phenotypically active compounds, even for cases where genetic approaches are not tractable. Drug discovery efforts based on phenotypic screens have successfully identified thousands of molecules that kill *Plasmodium* parasites (*Plouffe et al., 2008*; *Gamo et al., 2010*; *Guiguemde et al., 2012*), but most of these hits act by unknown mechanisms. Combined, these complementary target identification strategies will provide greater coverage of the druggable proteome and maximise the chance of deconvoluting targets of compounds with uncharacterised modes of action. These strategies can assist in the discovery of new antimalarial drug targets that will facilitate the development of next-generation medicines with novel and diverse mechanisms, ensuring effective treatments for malaria are available in the future.

## Methods
### Chemistry
#### Synthetic materials and methods

Chemicals and solvents were purchased from standard suppliers and used without further purification. $^1$H NMR, $^{13}$C NMR, and $^{19}$F NMR spectra were recorded on a Bruker Avance Nanobay III 400MHz Ultrashield Plus spectrometer at 400.13 MHz, 100.61 MHz, and 376.50 MHz, respectively. Chemical shifts ($\delta$) are recorded in parts per million (ppm) with reference to the chemical shift of the deuterated solvent. Unless otherwise stated, samples were dissolved in CDCl$_3$. Coupling constants ($J$) and carbon-fluorine coupling constants ($J_{CF}$) are recorded in Hz and multiplicities are described by singlet (s), doublet (d), triplet (t), quadruplet (q), broad (br), multiplet (m), doublet of

doublets (dd), doublet of triplets (dt). LC-MS was performed using either system A or B. System A: an Agilent 6100 Series Single Quad coupled to an Agilent 1200 Series HPLC using a Phenomenex Luna C8 (2) 50×4.6 mm, 5 µm column. The following buffers were used: buffer A: 0.1% formic acid (FA) in $H_2O$; buffer B: 0.1% FA in MeCN. Samples were run at a flow rate of 0.5 mL/min for 10 min: 0–4 min 5–100% buffer B in buffer A, 4–7 min 100% buffer B, 7–9 min 100–5% buffer B in buffer A, 9–10 min 5% buffer B in buffer A. Mass spectra were acquired in positive and negative ion mode with a scan range of 100–1000 $m/z$. UV detection was carried out at 254 nm. System B: an Agilent 6120 Series Single Quad coupled to an Agilent 1260 Series HPLC using a Poroshell 120 EC-C18 50×3.0 mm, 2.7 µm column. The following buffers were used: buffer A: 0.1% FA in $H_2O$; buffer B: 0.1% FA in MeCN. Samples were run at a flow rate of 0.5 mL/min for 5 min; 0–1 min 5% buffer B in buffer A, 1–2.5 min 5–100% buffer B in buffer A, 2.5–3.8 min 100% buffer B, 3.8–4 min 100–5% buffer B in buffer A, 4–5 min 5% buffer B in buffer A. Mass spectra were acquired in positive and negative ion mode with a scan range of 100–1000 $m/z$. UV detection was carried out at 214 nm and 254 nm.

*Methyl 2-(2-hydroxy-2-methylpropanamido)-2-(3',4',5'-trifluoro-[1,1'-biphenyl]-4-yl)acetate (**3**)*. A mixture of methyl 2-amino-2-(4-bromophenyl)acetate (**2**) (400 mg, 1.64 mmol), 2-hydroxy-2-methylpropanoic acid (205 mg, 1.97 mmol), EDCI (1.2 eq) and DMAP (1.3 eq) in DCM (10 mL/mmol of amine) was stirred at room temperature under nitrogen overnight. The mixture was diluted with DCM, washed with 2 M $HCl_{(aq)}$, saturated $NaHCO_{3(aq)}$, then brine. The organic layer was dried over $Na_2SO_4$, filtered and concentrated under reduced pressure. The product was subsequently reacted with 3,4,5-trifluorophenylboronic acid (347 mg, 1.97 mmol), $Pd(PPh_3)_2Cl_2$ (35 mg, 0.05 mmol), $Na_2CO_3$ (1 M, 3.3 mL) in THF (9.9 mL) in a sealed microwave vial and heated at 100°C for 2 hr. After cooling, the mixture was diluted with EtOAc (10 mL) and water (10 mL), and the mixture extracted with EtOAc. The organic layer was dried over $Na_2SO_4$, filtered and concentrated. The crude product was purified by column chromatography to give compound **3** (503 mg, 80%) as a colourless oil. $^1H$ NMR δ 7.73 (br. d, *J*=7.0 Hz, 1H), 7.53–7.39 (m, 4H), 7.22–7.06 (m, 2H), 5.57 (d, *J*=7.3 Hz, 1H), 3.76 (s, 3H), 1.50 (s, 3H), 1.44 (s, 3H); $^{19}F$ NMR δ –133.85 (d, *J*=20.5 Hz), –162.09 (dd, *J*=20.5 Hz); $^{13}C$ NMR δ 175.9, 171.1, 151.6 (ddd, $J_{CF}$ = 250.3/10.2/4.6 Hz), 141.2–138.0 (m, 2C), 136.9, 136.8–136.5 (m), 128.1, 127.7, 111.6–110.9 (m), 73.8, 56.1, 53.1, 28.0, 27.9; LC-MS $t_R$: 3.4 min, $m/z$ 381.9 [MH]+.

*2-Hydroxy-N-(2-(hydroxyamino)-2-oxo-1-(3',4',5'-trifluoro-[1,1'-biphenyl]-4-yl)ethyl)-2-methylpropanamide [MIPS2673 (**4**)]*. Methyl 2-(2-hydroxy-2-methylpropanamido)-2-(3',4',5'-trifluoro-[1,1'-biphenyl]-4-yl)acetate (**3**) (100 mg, 0.26 mmol) was dissolved in anhydrous MeOH (5 mL/mmol of ester) at room temperature. $NH_2OH·HCl$ (4.0 eq) was added followed by KOH (5 M in anhydrous MeOH, 5.0 eq). The mixture was stirred at room temperature overnight and monitored by LC-MS analysis. The mixtures were directly dry-loaded on to Isolute HM-N (Biotage), before purification by FCC. The desired hydroxamic acid 3 was obtained as white solid (40 mg, 40% yield). $^1H$ NMR (DMSO-$d_6$) δ 11.13 (s, 1H), 9.15 (s, 1H), 8.07 (d, *J*=8.1 Hz, 1H), 7.78–7.63 (m, 4H), 7.46 (d, *J*=8.3 Hz, 2H), 5.74 (s, 1H), 5.29 (d, *J*=8.1 Hz, 1H), 1.26 (s, 3H), 1.23 (s, 3H); $^{19}F$ NMR (DMSO-$d_6$) δ –134.91 (d, *J*=21.7 Hz), –163.45 (dd, *J*=21.8 Hz). $^{13}C$ NMR (DMSO-$d_6$) δ 176.2, 166.4, 150.9 (ddd, $J_{CF}$ = /9.8/4.3 Hz), 139.5, 140.2–137.1 (m), 136.9–136.4 (m, 2C), 127.3, 127.2, 111.8–110.9 (m), 72.5, 53.2, 27.8, 27.6; $m/z$ HRMS (TOF ES+) $C_{18}H_{18}F_3N_2O_4$ [MH]+ calcd 383.1213; found 383.1214; LC-MS $t_R$: 3.2 min; HPLC $t_R$: 7.7 min, >99%.

## Routine parasite culture

*P. falciparum* parasites (3D7 line) were cultured continuously in O+ human red blood cells (Australian Red Cross Blood Service) at 2–4% haematocrit using standard methods (*Trager and Jensen, 1976*), with minor modifications (*Creek et al., 2016*). Briefly, cultures were maintained at 37°C under defined atmospheric conditions (94% $N_2$, 5% $CO_2$, and 1% $O_2$) in RPMI 1640 medium supplemented with 0.5% Albumax II (Gibco, Australia), HEPES (5.94 g/L), hypoxanthine (50 mg/L), and sodium bicarbonate (2.1 g/L). Parasites were synchronised using multiple rounds of sorbitol lysis (*Lambros and Vanderberg, 1979*) and monitored at least every 48 hr by Giemsa staining of methanol-fixed blood smears.

Parasite material for thermal stability and limited proteolysis experiments was obtained from cultures synchronised to 30–38 hr post invasion. Parasites were isolated from red blood cells via hypotonic lysis to avoid the use of solubilising molecules such as saponin. Briefly, infected red blood cells were magnetically purified and incubated with 10 mL of hypotonic lysis buffer (150 mM ammonium chloride, 10 mM potassium bicarbonate, 1 mM EDTA in Milli-Q water) for 5 min at 4°C. Following

hypotonic lysis, the intact parasite pellets were washed three times in phosphate-buffered saline (PBS) and stored at –80°C until whole proteome preparation.

## Whole proteome preparation

Parasite pellets were resuspended in 1–2 mL of cold 100 mM HEPES buffer (pH 8.1) and subjected to freeze-thaw lysis by cycling at 2–3 min intervals between dry ice and a 35°C heat block. Cells used for limited proteolysis also underwent mechanical shearing after freeze-thaw lysis by passing the thawed suspension through a 26-gauge (10–20×) and then a 30.5-gauge needle (10–20×). The resulting lysates were centrifuged to remove cell debris (20,000×$g$ for 20 min at 4°C), the supernatants collected and combined to create one pooled parasite lysate. For thermal stability experiments, the cell debris obtained after centrifugation underwent two additional cycles of freeze-thaw and the collected supernatants were added to the pooled lysate. All remaining steps were performed at 4°C. The protein concentration of the resulting lysate was determined using the Pierce bicinchoninic acid (BCA) protein assay kit (Thermo Fisher Scientific).

## Lysate treatment for thermal stability proteomics and mass spectrometry analysis

The pooled lysate was equally separated to achieve at least three incubations (technical replicates) per condition. The parasite lysates were incubated with 1 µM or 4 µM of MIPS2673 or vehicle (DMSO) for 3 min at room temperature, followed by thermal challenge at 60°C for 5 min. Each sample was incubated at room temperature for a further 3 min prior to being returned to ice. The denatured protein was removed via ultracentrifugation at 100,000×$g$ for 20 min (4°C) in a Beckman Coulter Optima XE-90 – IVD ultracentrifuge with a 42.2 Ti rotor. The protein concentration in the soluble fraction was determined using a BCA assay. Samples were then reduced and alkylated at 95°C for 5 min using 10 mM tris(2-carboxyethyl)phosphine hydrochloride (TCEP) and 40 mM iodoacetamide (40 mM final concentration), respectively. Each sample was digested overnight with mass spectrometry-grade trypsin (1:50; Promega) at 37°C while shaking. The following day, trypsin activity was quenched using 5% FA and samples were subjected to desalting using in-house-generated StageTips, as described previously (*Rappsilber et al., 2003*). The samples were then dried and resuspended in 12 µL of 2% (vol/vol) acetonitrile (ACN) and 0.1% (vol/vol) FA containing indexed retention time (iRT) peptides (Biognosys) for LC-MS/MS analysis.

## Lysate treatment for limited proteolysis and mass spectrometry analysis

An equivalent volume of protein lysate was aliquoted from a lysate pool for a minimum of three independent incubations per condition and incubated with MIPS2673 (0.1–10 µM) or vehicle (DMSO) at room temperature for 10 min. Proteinase K from *Tritirachium album* (Sigma-Aldrich) was added to all of the reactions at a 1:100 ratio of enzyme to protein and incubated at room temperature for 4 min. The digestion reactions were stopped by heating samples to 98°C for 1 min, followed by addition of an equal volume of 10% sodium deoxycholate (Sigma-Aldrich) to achieve a final concentration of 5%. Samples were incubated for a further 15 min at 98°C. These samples were then removed from the heat and subjected to complete digestion under denaturing conditions.

Samples were reduced and alkylated for 10 min at 95°C with 10 mM TCEP and 40 mM chloroacetamide, respectively. Subsequently, samples were digested with Trypsin/LysC (1:100 enzyme to substrate ratio) for 16 hr at 37°C while shaking. Deoxycholate was precipitated by addition of FA to a final concentration of 1.5% and centrifuged at 16,000×$g$ for 10 min. After transferring the supernatant to a new tube an equal volume of FA was added again and the centrifugation repeated. Digests were desalted using in-house-generated StageTips (*Rappsilber et al., 2003*) and after drying resuspended in 12 µL of 2% (vol/vol) ACN and 0.1% (vol/vol) FA containing iRT peptides (Biognosys) for LC-MS/MS analysis.

## LC-MS/MS acquisition for proteomics

LC-MS/MS was carried out using data-independent acquisition mode on a Q-Exactive HF mass spectrometer (Thermo Scientific) as described previously (*Siddiqui et al., 2022a*). Briefly, samples were loaded at a flow rate of 15 µL/min onto a reversed-phase trap column (100 µm × 2 cm), Acclaim

PepMap media (Dionex), maintained at a temperature of 40°C. Peptides were eluted from the trap column at a flow rate of 0.25 µL/min through a reversed-phase capillary column (75 µm × 50 cm) (LC Packings, Dionex). The HPLC gradient was 158 min and gradually reached 30% ACN after 123 min, 34% ACN after 126 min, 79% ACN after 131 min, and 2% after 138 min for a further 20 min. The mass spectrometer was operated in a data-independent mode with a 43-fixed-window setup of 18 $m/z$ effective precursor isolation over the $m/z$ range of 364–988 Da. Full scan was performed at 60,000 resolution (AGC target of $3e^6$ and maximum injection time of 54 ms) with fragmentation resolution at 15,000 (AGC target of $2e^5$, maximum injection time of 22 ms, normalised collision energy of 27.0).

## Data analysis for thermal stability proteomics experiments

Raw files were processed using Spectronaut 13.0 against an in-house-generated *P. falciparum* infected red blood cell spectral library as described previously (*Siddiqui et al., 2022a*). The consensus library contained 42,245 peptides corresponding to 4421 protein groups. For processing, raw files were loaded and Spectronaut calculated the ideal mass tolerances for data extraction and scoring based on its extensive mass calibration with a correction factor of 1. Both at precursor and fragment level, the highest data point within the selected $m/z$ tolerance was chosen. Identification of peptides against the library was based on default Spectronaut settings (Manual for Spectronaut 13.0, available on Biognosis website). Briefly, precursor Qvalue Cut-off and Protein Qvalue Cut-off were as per default at 1%, therefore only those that passed the cut-off were considered as identified and used for subsequent processing. Retention time prediction type was set to dynamic iRT. Interference correction was on MS2 level. For quantification, the interference correction was activated and a cross-run normalisation was performed using the total peak area as the normalisation base with a significance level of 0.01.

Significantly stabilised proteins were determined by first calculating the fold-change between the MIPS2673 (1 µM and 4 µM conditions) and vehicle-treated samples for each experiment. Statistical significance of the change in protein abundance was determined by a Welch's t-test. Proteins that were significantly stabilised (fold-change >1.2, p<0.05) at both MIPS2673 concentrations in multiple experiments were considered putative targets. The data were plotted using paired volcano plots, whereby fold-change in protein abundance was plotted as a function of statistical significance for multiple conditions or experiments to visualise commonly stabilised proteins.

## Data analysis for limited proteolysis experiments

For limited proteolysis experiments, raw data files were processed using Spectronaut 13.0 as described above, except a limited proteolysis-specific *P. falciparum* infected red blood cell spectral library was used. The in-house-generated library contained 85,039 peptides corresponding to 4681 protein groups.

All limited proteolysis datasets were analysed at the modified peptide sequence level. The datasets were first analysed for differentially abundant peptides between MIPS2673 and vehicle by filtering based upon relative peptide abundance (absolute fold-change >1.5) and statistical significance (q<0.01). Within independent experiments, peptides meeting these thresholds at each drug concentration were considered significant. To filter and prioritise possible protein targets, a one-sided Fisher's exact test was used to determine whether, based on the total number of peptides detected for each protein in the dataset, the number of significant LiP peptides identified for each protein was greater than would be expected by chance. Resulting p-values were adjusted for multiple testing using the Bonferroni method. Proteins passing the Bonferroni corrected statistical cut-off were considered putative targets. Proteins that did not pass the stringent Bonferroni corrected statistical cut-off, but were statistically significant (p<0.05), were considered lower confidence targets.

The position of drug binding using the centre of mass and distance measurements were performed using PyMol 2.5.1 (Schrodinger). Distance measurement analyses considered only peptides detected in both LiP-MS experiments. For the MIPS2673-*Pf*A-M1 interaction, we mapped peptides to our experimentally determined *Pf*A-M1 structure with MIPS2673 bound (PDB: 8SLO). Significant LiP peptides were defined as peptides that passed the relative abundance (absolute fold-change >1.5), statistical significance (q<0.01), and proteolytic peptide pattern (i.e. increased fully tryptic or decreased half tryptic) filters at all concentrations tested in both experiments. For other putative MIPS2673-interacting proteins identified by thermal stability proteomics and by LiP-MS in experiment one, only one significant LiP peptide was identified (YSPSFMSFK from *Pf*ADA)

using the above criteria to define significant LiP peptides. Thus, for distance measurement analyses of these other putative MIPS2673-interacting proteins, we applied a less stringent rule for defining significant LiP peptides, namely, any peptide that passed the relative abundance (absolute fold-change >1.5), statistical significance (q<0.01), and proteolytic peptide pattern (i.e. increased fully tryptic or decreased half tryptic) filters at any concentration in both experiments. The active site metal ion (*Pf*ADA PDB: 6II7, *Pf*A-M17 PDB: 7RIE, *Pf*APP PDB: 5JR6) or ligand (*Pf*MIF PDB: 4P7S, TRXR1 PDB: 2ZZC) were used as a reference point for minimum distance measurements. For RAB39A, which is not a metalloprotein and where no ligand-bound structure is available, the GTP binding site residues were used (AlphaFold: AF-Q14964-F1). The putative *Pf*ApiAP2 transcription factor, PF3D7_1239200, was excluded from this analysis as no protein structure is available and the AlphaFold predicted structure is of very low quality (pLDDT score 36.67). No peptides met the significance criteria for *Pf*A-M18 and the conserved parasite proteins, PF3D7_1026000 and PF3D7_0604300.

To reveal in an unbiased way whether any protein functions are overrepresented in the expanded list of candidate targets from the experiment one dataset, this list was tested for functional enrichments using the *top*GO R-package (*Moldovan et al., 2020*). All proteins with an uncorrected p<0.05 were tested against a background of proteins for which abundance was measured. PlasmoDB GO terms were used for GO term mapping. We tested for enrichment in molecular function GO terms with the Fisher's exact test using the weight-algorithm in topGO (*Alexa et al., 2006*). All terms with a p-value less than 0.05 were considered significant. GO terms represented by fewer than five proteins in the background dataset were excluded from the analysis.

## Enzyme assays

Inhibition of aminopeptidase activity assays were performed against nine aminopeptidases: the M1 alanyl and M17 leucyl aminopeptidases from *P. falciparum* (*Pf*A-M1; *Pf*A-M17) and *P. vivax* (*Pv*A-M1; *Pv*A-M17), the M18 aspartyl and aminopeptidase P aminopeptidases from *P. falciparum* (*Pf*A-M18; *Pf*APP) and three human M1 homologues: LTA4H (OriGene TP307617), ERAP1 (OriGene TP314469), and ERAP2 (Creative BioMart ERAP2-304H). The *Plasmodium* enzymes were produced recombinantly as described previously (*McGowan et al., 2009*; *McGowan et al., 2010*; *Malcolm et al., 2021*; *Drinkwater et al., 2016a*; *Sivaraman et al., 2012*) and the human recombinant enzymes were purchased from commercial suppliers as indicated.

The ability of MIPS2673 to inhibit aminopeptidase activity was assessed by fluorescence assays using fluorogenic substrates L-leucine-7-amido-4-methylcoumarin hydrochloride (H-Leu-NHMec, 20 µM for *Pf*A-M1, 40 µM for *Pv*A-M1, and 100 µM for ERAP1 and ERAP2) for all enzymes except LTA4H that was assessed using L-alanine-7-amido-4-methylcoumarin hydrochloride (H-Ala-NHMec, 20 µM) and *Pf*APP that was assessed using Lys(Abz)-Pro-Pro-NA (PPpNA, 10 µM) (*Drinkwater et al., 2016b*). Aminopeptidase activity of *Pf*-M18 was measured by an absorbance assay using the substrate L-glutamic acid γ-(4-nitroanilide) (L-Glu-pNA, 100 µM) (*Sivaraman et al., 2012*). All assays were performed at pH 8.0. The concentration of substrate in the assay was held constant for each enzyme and ranged from 10 µM to 100 µM depending on the enzyme. Reactions were measured at 37°C in white 384-well plates in a total volume of 50 µL using a spectrofluorimeter (BMG FLUOstar) with excitation at 355 nm and emission at 460 nm for the fluorescence assays or excitation at 405 nm for the absorbance assays. The fluorescence or absorbance signal was continuously monitored until a final steady-state velocity, v, was obtained. Inhibition constants were calculated in biological triplicate from three different protein preparations. For determination of the apparent Morrison inhibition constant (denoted here as $K_i$), enzymes were pre-incubated in 100 mM Tris-HCl, pH 8.0 (supplemented with 1 mM $CoCl_2$ for M17 aminopeptidases) and inhibitor added 20 min prior to the addition of substrate. Substrate concentrations were selected to allow sensitive detection of enzyme activity while not exceeding the $K_m$ for each enzyme. A compound concentration range was selected to obtain a complete inhibition curve (0–100%) in biological triplicate. Where possible, the $K_i$ values were calculated by plotting the initial rates versus inhibitor concentration, and fitting to the Morrison equation for tight binding inhibitors in GraphPad Prism (non-linear regression method). Where a $K_i$ could not be calculated (ERAP1), the percentage inhibition was calculated assuming 100% activity in the absence of compound.

## Structural biology

*Pf*A-M1 was co-crystallised with MIPS2673 by the hanging-drop method using previously established protocols (*McGowan et al., 2009*). Purified protein was concentrated to 5.0 mg/mL and mixed with MIPS2673 to a final ligand concentration of 1 mM. Crystals grew in 20–30% poly(ethylene glycol) (PEG)8000, 0.1 M Tris pH 7.5–8.5, 0.2 M $MgCl_2$, 10% glycerol. Co-crystals were subjected to an additional overnight compound soak before being snap-frozen in liquid nitrogen. Data were collected at 100 K using synchrotron radiation at the Australian Synchrotron beamlines 3BM1 (*Cowieson et al., 2015*). Data were processed using XDS (*Kabsch, 2010*) and Aimless (*Evans and Murshudov, 2013*) as part of the CCP4i program suite (*Winn et al., 2011*). The structures were solved by molecular replacement in Phaser (*McCoy et al., 2007*) using the structure of unliganded *Pf*A-M1 coordinates (RCSB ID 3EBG) as the search models. The structures were refined using Phenix (*Adams et al., 2010*), with 5% of reflections set aside for calculation of $R_{free}$. Between refinement cycles, the protein structure, solvent, and inhibitors were manually built into $2F_o – F_c$ and $F_o – F_c$ electron density maps using COOT (*Emsley and Cowtan, 2004*; *Emsley et al., 2010*), with restraint files generated by Phenix where necessary. The data collection and refinement statistics can be found in Supplemental Information (*Figure 1—source data 1*). The coordinates and structure factors are available from the Protein Data Bank with PDB Accession codes: 8SLO.pdb (*Pf*A-M1-MIPS2673).

## Determination of compound $EC_{50}$ against 3D7 *P. falciparum* parasites

Parasite viability assay was performed as previously described (*Edgar et al., 2022*). Synchronised ring stage *P. falciparum* 3D7 parasites were cultured in 96-well U-bottom plates at 0.5% parasitaemia and 2% haematocrit, to which 50 µL of serially diluted MIPS2673 or artesunate was added. After 72 hr plates were placed at –80°C before being analysed in buffer containing 20 mM Tris pH 7.5, 5 mM EDTA, 0.008% saponin (wt/vol), and 0.008% Triton X-100 (vol/vol) (*Smilkstein et al., 2004*) containing 0.2 µL/mL SYBR Green I Nucleic Acid Gel Stain (10,000× in DMSO; Thermo Fisher). After 1 hr incubation, fluorescence intensity was read on a Glomax Explorer Fully Loaded plate reader (Promega) at emission wavelengths of 500–550 nm and an excitation wavelength of 475 nm. Uninfected RBCs and parasites treated with DMSO were used to normalise fluorescence. Data from four biological replicates performed in triplicate were plotted as 4-parameter log dose non-linear regression analysis with a sigmoidal dose-response curve fitted using GraphPad Prism 9 to generate $EC_{50}$ values, with error bars representative of the SEM.

## $EC_{50}$ determination against drug-resistant field isolates and laboratory selected *P. falciparum* parasites

In vitro testing was performed with a modified [3H]-hypoxanthine incorporation assay, as previously reported (*Snyder et al., 2007*).

## Parasite killing rate assay

Assay was performed as previously described (*Edgar et al., 2022*). Synchronised ring or trophozoite *P. falciparum* 3D7 parasites were cultured in the presence of 10× the $EC_{50}$ of MIPS2673 or artesunate for either 24 hr or 48 hr. Media containing compound was replenished at 24 hr for 48 hr assays. After incubation, RBCs were thoroughly washed to remove the compound and cultures were diluted 1/3 with fresh media and further grown for 48 hr before being placed at –80°C. Cultures were thawed and analysed using SYBR Green I assay as described above. Parasite viability was determined as a percentage of DMSO-treated parasites cultured alongside compound-treated parasites. Artesunate was used as a comparison of the parasite killing rate, and experiments were performed in four biological replicates.

## Gametocyte activity assay

Viability assays were performed as previously described (*Duffy and Avery, 2013*; *Duffy et al., 2016*; *Lucantoni et al., 2017*) against *Pf*NF54-*s16*-GFP early (I-III); late (IV-V), and mature (V) stage gametocytes. Gametocytes were assessed at early stage (day 2), late stage (day 8), and mature (day 12). Compounds diluted in 4% DMSO were transferred into 384-well imaging plates; gametocytes at various stages added, and plates incubated for 72 hr in 5% $CO_2$, 5% $O_2$, and 60% humidity at 37°C. After 72 hr incubation, 5 µL of MitoTracker Red CMH2XRos in PBS added per well and plates incubated overnight. Image acquisition and analysis was undertaken on the Opera QEHS micro-plate

confocal imaging system. An Acapella-based script using the MTR fluorescent signal and the GFP designated object quantifying viable stage-dependent parasite morphology identified gametocytes. Gametocyte viability was calculated as a percentage of the positive (5 µM puromycin) and negative (0.4% DMSO) controls. $EC_{50}$ values were calculated using a 4-parameter log dose, non-linear regression analysis, with sigmoidal dose-response (variable slope) curve fit using GraphPad Prism (version 4.0). No constraints were used in the curve fit. Chloroquine, artesunate, pyronaridine, pyrimethamine, dihydroartemisinin, and methylene blue were analysed in parallel as reference compounds. Experiments were completed in duplicate for two or three biological replicates.

### HEK293 cytotoxicity assay

To assess the cytotoxicity of MIPS2673 against mammalian cells, an established and well-validated metabolic assay for HEK293 cells was used (*Duffy and Avery, 2012*). In brief, HEK293 cells were maintained in DMEM supplemented with 10% fetal bovine serum. Compound testing was undertaken in tissue culture-treated 384-well plates incubated for 72 hr at 37°C and 5% $CO_2$. The media was then removed and replaced with resazurin (final concentration 44 µM). After an additional 6 hr incubation, total fluorescence was measured using an Envision plate reader (Perkin Elmer) (excitation/emission: 530 nm/595 nm).

### Metabolomics

*P. falciparum* 3D7 cultures that underwent double sorbitol synchronisation were incubated for a further 28–42 hr to achieve the desired trophozoite stage (28 hr post invasion). Infected RBC cultures at 6% parasitaemia and 2% haematocrit ($2\times10^8$ cells in total) were treated with 3× the $EC_{50}$ of MIPS2673 (1 µM) or an equivalent volume of DMSO (negative control) for 1 hr, after which metabolites were extracted. Following incubation, all samples were centrifuged at 650×*g* for 3 min, supernatants were removed, and pellets were washed in 1 mL of ice-cold PBS. Samples were again centrifuged at 650×*g* for 3 min to remove all of the PBS and the pellets were resuspended in 150 µL of ice-cold extraction buffer (100% methanol). Samples were then incubated on an automatic vortex mixer for 1 hr at 4°C before being centrifuged at 21,000×*g* for 10 min. Supernatant was collected and 100 µL was stored at –80°C until further analysis and 10 µL from each sample was pooled to serve as a quality control sample. LC-MS analysis and data processing were as previously described (*Edgar et al., 2022*).

### Kinetic solubility estimation

Kinetic solubility was estimated using nephelometry ($Sol_{pH}$) according to the method of *Bevan and Lloyd, 2000*. Briefly, the compound in DMSO was spiked into either pH 6.5 phosphate buffer or 0.01 M HCl (approximately pH 2) with the final DMSO concentration being 1%. After 30 min had elapsed, samples were then analysed via nephelometry to determine a solubility range.

## Acknowledgements

The authors wish to thank and acknowledge the Australian Red Cross LifeBlood for the provision of fresh red blood cells, without which antimalarial testing could not have been performed. The Centre for Drug Candidate Optimisation (CDCO, Monash University) is acknowledged for providing the solubility data. The CDCO is partially supported by the Monash University Technology Research Platform network and Therapeutic Innovation Australia (TIA) through the Australian Government National Collaborative Research Infrastructure Strategy (NCRIS) program. We thank the following colleague for support with the *P. falciparum* strain panel assays: Sibylle Sax, Swiss Tropical, and Public Health Institute, Switzerland. We thank the Monash Proteomics and Metabolomics Facility for technical support. SD was supported by a NHMRC Dora Lush Biomedical Postgraduate Scholarship (APP1150359), Griffith University DVCR Postgraduate Top-up Scholarship, and a Discovery Biology Top-up Scholarship. VMA acknowledges Medicines for Malaria Venture for continued financial support. Funding was provided by an NHMRC Synergy Grant for the Antimalarial Synergy Team #1185354.

## Additional information

### Funding

| Funder | Grant reference number | Author |
|---|---|---|
| National Health and Medical Research Council | APP1185354 | Tania De Koning-Ward Sheena McGowan Darren J Creek |
| National Health and Medical Research Council | APP1150359 | Sandra Duffy |
| Medicines for Malaria Venture | | Vicky M Avery |

The funders had no role in study design, data collection and interpretation, or the decision to submit the work for publication.

### Author contributions

Carlo Giannangelo, Conceptualization, Data curation, Formal analysis, Investigation, Visualization, Methodology, Writing – original draft, Project administration, Writing – review and editing; Matthew P Challis, Conceptualization, Data curation, Formal analysis, Investigation, Visualization, Methodology, Writing – review and editing; Ghizal Siddiqui, Data curation, Formal analysis, Investigation, Visualization, Writing – original draft, Writing – review and editing; Rebecca Edgar, Data curation, Formal analysis, Investigation, Visualization, Writing – review and editing; Tess R Malcolm, Nyssa Drinkwater, Sandra Duffy, Data curation, Formal analysis, Investigation; Chaille T Webb, Data curation, Formal analysis, Investigation, Writing – review and editing; Natalie Vinh, Data curation, Investigation, Writing – review and editing; Christopher Macraild, Formal analysis, Visualization, Writing – review and editing; Natalie Counihan, Investigation; Sergio Wittlin, Resources, Data curation, Formal analysis; Shane M Devine, Supervision, Writing – review and editing; Vicky M Avery, Resources, Supervision, Writing – review and editing; Tania De Koning-Ward, Conceptualization, Resources, Supervision, Funding acquisition, Project administration, Writing – review and editing; Peter Scammells, Conceptualization, Resources, Funding acquisition, Visualization, Writing – original draft, Project administration, Writing – review and editing; Sheena McGowan, Conceptualization, Resources, Formal analysis, Supervision, Funding acquisition, Visualization, Writing – original draft, Project administration, Writing – review and editing; Darren J Creek, Conceptualization, Resources, Supervision, Funding acquisition, Writing – original draft, Project administration, Writing – review and editing

### Author ORCIDs

Carlo Giannangelo ⓘ https://orcid.org/0000-0002-6654-7266
Natalie Counihan ⓘ https://orcid.org/0000-0002-8973-3344
Shane M Devine ⓘ https://orcid.org/0000-0001-8727-7392
Sheena McGowan ⓘ https://orcid.org/0000-0001-6863-1106
Darren J Creek ⓘ https://orcid.org/0000-0001-7497-7082

Reviewer #1 (Public Review): https://doi.org/10.7554/eLife.92990.3.sa1
Reviewer #2 (Public Review): https://doi.org/10.7554/eLife.92990.3.sa2
Reviewer #3 (Public Review): https://doi.org/10.7554/eLife.92990.3.sa3
Author response https://doi.org/10.7554/eLife.92990.3.sa4

## Additional files

### Supplementary files
• MDAR checklist

### Data availability

The mass spectrometry proteomics data have been deposited to the ProteomeXchange Consortium via the PRIDE (*Perez-Riverol et al., 2022*) partner repository (https://www.ebi.ac.uk/pride/archive/)

with the dataset identifier PXD044125.The mass spectrometry metabolomics data is available at the NIH Common Fund's National Metabolomics Data Repository (NMDR) website, the Metabolomics Workbench, https://www.metabolomicsworkbench.org where it has been assigned Project ID (ST002792), the data can be accessed directly via it's Project DOI: 10.21228/M8SQ7X. NMDR is supported by Metabolomics Workbench/National Metabolomics Data Repository (NMDR) (grant# U2C-DK119886), Common Fund Data Ecosystem (CFDE) (grant# 3OT2OD030544) and Metabolomics Consortium Coordinating Center (M3C) (grant# 1U2C-DK119889). Structural data has been deposited with PDB with the identifier 8SLO. This article does not report original code. Any additional information required to reanalyze the data reported in this article is available from the lead contact upon request.

The following datasets were generated:

| Author(s) | Year | Dataset title | Dataset URL | Database and Identifier |
|---|---|---|---|---|
| Ghizal S | 2023 | Chemoproteomics validates selective targeting of Plasmodium M1 alanyl aminopeptidase as a cross-species strategy to treat malaria | https://doi.org/10.21228/M8SQ7X | Metabolomics Workbench, 10.21228/M8SQ7X |
| Giannangelo C, Creek DJ | 2024 | Chemoproteomics validates selective targeting of Plasmodium M1 alanyl aminopeptidase as a cross-species strategy to treat malaria | https://www.ebi.ac.uk/pride/archive/projects/PXD044125 | PRIDE, PXD044125 |
| McGowan SM, Drinkwater N | 2024 | *Plasmodium falciparum* M1 aminopeptidase bound to selective inhibitor MIPS2673 | https://www.rcsb.org/structure/8SLO | RCSB Protein Data Bank, 8SLO |

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
