## [Editor Report · eLife assessment]

The manuscript makes an **important** contribution to antimalarial drug discovery, utilizing diverse systems biology methodologies. It focuses on an improved M1 metalloprotease inhibitor and provides **compelling** evidence for the utility of chemoproteomics in pinpointing PfA-M1 targeting. Additionally, metabolomic analysis reveals specific alterations in the final steps of hemoglobin breakdown. These findings highlight the potential of the developed methodology not only for PfA-M1 targeting but also for other inhibitors targeting various malarial proteins or pathways.

---

## [Referee Report · Reviewer #1 (Public Review)]

By using a series of biochemical methods based on proteomic and metabolomic approaches, this study aims at: (1) validating the specific targeting of a biologically active molecule (MIPS2673) towards a defined (and unique?) protein target within a parasite, and (2) exploring whether it is possible to extrapolate which metabolic pathway has been disrupted.

Strength/Weaknesses

-The chemoproteomic approach, convincingly shows that MIPS2673 more significantly "protects" the putative target (PfA-M1) against thermal degradation or against enzymatic attack (by proteinase K). Proteomic studies are carried using parasite extracts enriched in late trophozoites (30-38 h pi), and are restricted to the soluble proteins fraction.

-The metabolomic approach, documents the ability of MIPS2673 to selectively increase the number of non-hydrolyzed dipeptides in treated versus untreated parasites, further arguing for selective targeting of PfA-M1 and impairment of hemoglobin breakdown by the parasite.

-The revised version now also considers and further studies the additional putative targets identified by one proteomic approach (but not the other one), which is both more critical of the results obtained and more realistic.

The work as a whole is highly interesting, both for the specific topic of PfA-M1's role in parasite biology and for the method, applicable to other malarial drug contexts.

---

## [Referee Report · Reviewer #2 (Public Review)]

In this manuscript, the authors first developed a new small molecular inhibitor that could target specifically the M1 metalloproteases of both important malaria parasite species Plasmodium falciparum and P. vivax. This was done by a chemical modification of a previously developed molecule that targets PfM1 as well as PfM17 and possibly other Plasmodial metalloproteases. After the successful chemical synthesis, the authors showed that the derived inhibitor, named MIPS2673, has a strong antiparasitic activity with IC50 342 nM and it is highly specific for M1. With this in mind, the authors first carried out two large-scale proteomics to confirm the MIPS2673 interaction with PfM1 in the context of the total *P. falciparum* protein lysate. This was done first by using thermal shift profiling and subsequently limited proteolysis. While the first demonstrated overall interaction, the latter (limited proteolysis) could map more specifically the site of MIPS2673-PfM1 interaction, presumably the active site. Subsequent metabolomics analysis showed that MIPS2673 cytotoxic inhibitory effect leads to the accumulation of short peptides many of which originate from hemoglobin. Based on that the authors argue that the MIPS2673 mode of action (MOA) involves inhibition of hemoglobin digestion that in turn inhibits the parasite growth and development.

Comments on the revised version:

The authors addressed all my comments from the previous round of reviews.

---

## [Referee Report · Reviewer #3 (Public Review)]

Summary:

Overall, this is an interesting series of experiments which have identified a putative inhibitor of the Plasmodium M1 alanyl aminopeptidases, PfA-M1 and PvA-M1. The weaknesses include the lack of additional analysis of additional targets identified in the chemoproteomic approaches.

Strengths:

The main strengths include the synthesis of MIPS2673 which is selectively active against the enzyme and in whole cell assay.

Weaknesses:

The authors have addressed the previously identified weaknesses and have now provided additional data and explanations. They have modified their conclusions to indicate the limitations of their work.

---

## [Author Response]

The following is the authors’ response to the original reviews.

**eLife assessment**
The manuscript constitutes an important contribution to antimalarial drug discovery, employing diverse systems biology methodologies; with a focus on an improved M1 metalloprotease inhibitor, the study provides convincing evidence of the utility of chemoproteomics in elucidating the preferential targeting of PfA-M1. Additionally, metabolomic analysis effectively documents specific alterations in the final steps of hemoglobin breakdown. These findings underscore the potential of the developed methodology, not only in understanding PfA-M1 targeting but also in its broader applicability to diverse malarial proteins or pathways. Revisions are needed to further enhance overall clarity and detail the scope of these implications.

We thank the editor and reviewers for recognising the contribution our work makes to understanding the selective targeting of aminopeptidase inhibitors in malaria parasites and the wider impact this multi-omic strategy can have for anti-parasitic drug discovery efforts. The reviewers have provided constructive feedback and raised important points that we have taken on-board to improve our manuscript. In particular, we have revised aspects of the text and figures to enhance clarity, performed additional analysis on the other possible MIPS2673 interacting proteins and more comprehensively analysed the effect of MIPS2673 on parasite morphology. NB: Specific responses to comments in the public reviews are provided within responses to the specific recommendations to authors.

**Public Reviews:**

**Reviewer #1 (Public Review):**
Summary:The article "Chemoproteomics validates selective targeting of Plasmodium M1 alanyl aminopeptidase as a cross-species strategy to treat malaria" presents a series of biochemical methods based on proteomics and metabolomics, as a means to:(1) validate the specific targeting of biologically active molecules (MIPS2673) towards a defined (unique) protein target within a parasite and (2) to explore whether by quantifying the perturbations generated at the level of the parasite metabolome, it is possible to extrapolate which metabolic pathway has been disrupted by using this biologically active molecule and whether this may further confirm selective targeting in parasites of the expected (or in-vitro targeted) enzyme (here PfA-1).The inhibitor used in this work by the authors (MIPS2673) is to my knowledge a novel one, although belonging to a chemical series previously explored by the authors, which recently enabled them to discover a specific PfA-M17 inhibitor, MIPS2571 (Edgard et al., 2022, ref 11 of this current work). Indeed, inhibitors specifically targeting either PfA-M1 or PfA-M17 (and not both, as currently done in the past) are scarce today, and highly needed to functionally characterize these two zinc-aminopeptidases. MIPS2673, blocks the development of erythrocytic stages of *Plasmodium falciparum* with an EC50 of 324 nM, blocks the parasite development at the young trophozoite stage at 5x EC50 (but at ring stages at 10xEC50, figure 1E), and inhibits the enzymatic activity of PfA-M1 (and its ortholog Pv-M1) but not of the related malarial metallo-aminopeptidases (M17 and M18 families) nor the human metalloenzymes from closely related enzymatic families, supporting its selective targeting of PfA-M1 (and Pv-M1).All experiments are carried out in vitro (e.g. biochemical studies such as enzymology, proteomics, metabolomics) and on cultured parasites (erythrocyte stages of *Plasmodium falciparum* and several gametocytes stages obtained in vitro); there are no in vivo manipulations. The work related to Plasmodium vivax, which justifies the "cross-species" indication in the title of the article, is restricted to using a recombinant form of the M1-family aminopeptidase in enzymatic assays. The rest of the work concerns only *Plasmodium falciparum*. While I found globally that this work is original and brings new data and above all proposes chemical validation approaches that could be used for other target validations under similar limiting conditions (impossibility of KO of the gene), I have some specific questions to address to the authors.Strengths and weaknesses:- The chemoproteomic approach, that explores the ability of MIPS2673 to more significantly "protect" the putative target (PfA-M1) against thermal degradation or enzymatic attack (by proteinase K), to document its selective targeting towards PfA-M1 (the inhibitor, once associated with its target, is expected to stabilize its structure or prevent the action of end proteases), uses several concentrations of MIPS2673 and provides convincing results. My main criticism is that these tests are carried out with parasite extracts enriched in 30-38 hours old forms, and restricted to the fraction of soluble proteins isolated from these parasitic forms, which still limits the scope of the analysis. It is clear that this methodological approach is a choice that can be argued both biologically (PfA-M1 is well expressed in these stages of the parasite development) and biochemically (it is difficult to do proteomic analyses on insoluble proteins) but I regret that the authors do not discuss these limitations further, notably, I would have expected (from Figure 1E) some targets to be also present at ring stages.- The metabolomic approach, by documenting the ability of MIPS2673 to selectively increase the number of non-hydrolyzed dipeptides in treated versus untreated parasites is another argument in favor of the selective targeting of PfA-M1 by MIPS2673, in particular by its broad-spectrum aminopeptidase action preferentially targeting peptides resulting from the degradation of hemoglobin by the parasite. The relative contribution of peptides derived from host hemoglobin versus other parasite proteins is, however, little discussed.The work as a whole remains highly interesting, both for the specific topic of PfA-M1's role in parasite biology and for the method, applicable to other malarial drug contexts.
**Reviewer #2 (Public Review):**
In this manuscript, the authors first developed a new small molecular inhibitor that could target specifically the M1 metalloproteases of both important malaria parasite species *Plasmodium falciparum* and P. vivax. This was done by a chemical modification of a previously developed molecule that targets PfM1 as well as PfM17 and possibly other Plasmodial metalloproteases. After the successful chemical synthesis, the authors showed that the derived inhibitor, named MIPS2673, has a strong antiparasitic activity with IC50 342 nM and it is highly specific for M1. With this in mind, the authors first carried out two large-scale proteomics to confirm the MIPS2673 interaction with PfM1 in the context of the total *P. falciparum* protein lysate. This was done first by using thermal shift profiling and subsequently limited proteolysis. While the first demonstrated overall interaction, the latter (limited proteolysis) could map more specifically the site of MIPS2673-PfM1 interaction, presumably the active site. Subsequent metabolomics analysis showed that MIPS2673 cytotoxic inhibitory effect leads to the accumulation of short peptides many of which originate from hemoglobin. Based on that the authors argue that the MIPS2673 mode of action (MOA) involves inhibition of hemoglobin digestion that in turn inhibits the parasite growth and development.
**Reviewer #3 (Public Review):**
This is a manuscript that attempts to validate Plasmodium M1 alanyl aminopeptidase as a target for antimalarial drug development. The authors provide evidence that MIPS2673 inhibits recombinant enzymes from both Pf and Pv and is selective over other proteases. There is in vitro antimalarial activity. Chemoproteomic experiments demonstrate selective targeting of the PfA-M1 protease.This is a continuation of previous work focused on designing inhibitors for aminopeptidases by a subset of these authors. Medicinal chemistry explorations resulted in the synthesis of MIPS2673 which has improved properties including potent inhibition of PfA-M1 and PvA-M1 with selectivity over a closed related peptidase. The compound also demonstrated selectivity over several human aminopeptidases and was not toxic to HEK293 cells at 40 uM. The activity against *P. falciparum* blood-stage parasites was about 300 nM.Thermal stability studies confirmed that PfA-M1 was a binding target, however, there were other proteins consistently identified in the thermal stability studies. This raises the question as to their potential role as additional targets of this inhibitor. The authors dismiss these because they are not metalloproteases, but further analysis is warranted. This is particularly important as the authors were not able to generate mutants using in vitro evolution of resistance strategies. This often indicates that the inhibitor has more than one target.The next set of experiments focused on a limited proteolysis approach. Again several proteins were identified as interacting with MIPS2673 including metalloproteases. The authors go on to analyze the LiP-MS data to identify the peptide from PfA-M1 which putatively interacts with MIPS2673. The authors are clearly focused on PfA-M1 as the target, but a further analysis of the other proteins identified by this method would be warranted and would provide evidence to either support or refute the authors' conclusions.The final set of experiments was an untargeted metabolomics analysis. They identified 97 peptides as significantly dysregulated after MIPS2673 treatment of infected cells and most of these peptides were derived from one of the hemoglobin chains. The accumulation of peptides was consistent with a block in hemoglobin digestion. This experiment does reveal a potential functional confirmation, but questions remain as to specificity.Overall, this is an interesting series of experiments that have identified a putative inhibitor of PfA-M1 and PvA-M1. The work would be significantly strengthened by structure-aided analysis. It is unclear why putative binding sites cannot be analyzed via specific mutagenesis of the recombinant enzyme.In the thermal stability and LiP -MS analysis, other proteins were consistently identified in addition to PfA-M1 and yet no additional analysis was undertaken to explore these as potential targets.The metabolomics experiments were potentially interesting, but without significant additional work including different lengths of treatment and different stages of the parasite, the conclusions drawn are overstated. Many treatments disrupt hemoglobin digestion - either directly or indirectly and from the data presented here it is premature to conclude that treatment with MIPS2673 directly inhibits hemoglobin digestion.Finally, the potency of this compound on parasites grown in vitro is 300 nM - this would need improvements in potency and demonstration of in vivo efficacy in the SCID mouse model to consider this a candidate for a drug.Summary:Overall, this is an interesting series of experiments that have identified a putative inhibitor of the Plasmodium M1 alanyl aminopeptidases, PfA-M1 and PvA-M1.Strengths:The main strengths include the synthesis of MIPS2673 which is selectively active against the enzymes and in whole-cell assay.Weaknesses:The weaknesses include the lack of additional analysis of additional targets identified in the chemoproteomic approaches.
**Recommendations for the authors:**

**Reviewer #1 (Recommendations For The Authors):**
Question 1. Line 737 (and elsewhere). Why are Plasmodium vivax orthologs of PfA-M1 and PfA-M17 called Pv-M1 and Pv-M17 and not PvA-M1 and PvA-M17, where A stands for Aminopeptidase? I would recommend changing the names if possible, although the mention of Pv-M1 and Pv-M17 is now current in the literature (which is kind of regrettable). See also Supplemental Table S1 where PfA-M1 is named Pf-M1.

Supplemental Table S1 was updated to PfA-M1. Nomenclature for the Plasmodium vivax aminopeptidase orthologs was amended to PvA-M1 and PvA-M17 as suggested by the reviewer.

Question 2. Figure 1. Observation of parasite culture slide smears in Figure 1E strongly suggests that an important target of MIPS2673 appears to be expressed at the ring stage or very young trophozoites, whereas the authors, in their proteomic and metabolomic analyses, performed studies focused on late trophozoites stages (30-38h post-invasion). This difference in the targeting of Plasmodium stages puzzles me and deserves some explanations from the authors, and is related to my question 3.

As the reviewer indicates, ring-stage parasite growth appears to be affected at high concentrations (5x and 10x EC50) of MIPS2673. Under these conditions, parasite growth appears to stall during late rings/early trophs at ~16-22 h post invasion when haemoglobin digestion is increasing and when one presumes PfA-M1 (the primary target of MIPS2673) is increasing in both expression and activity (see references 26 and 28 of this manuscript). Thus, whilst it is unsurprising that MIPS2673 has some activity against ring-stage parasites, we focused on the trophozoite stage for our proteomics studies as we showed this to be the stage most susceptible to MIPS2673 (Fig. 1D) and reasoned that we would most likely identify the primary MIPS2673 target, and other interacting proteins, from a complex biological mixture at this stage. The same reasoning underpinned our decision to perform metabolomics on drug-treated trophozoites, as we reasoned we would see a greater functional effect on this stage. Furthermore, performing these experiments on trophozoites rather than rings minimises the interference from the host red blood cell. While we cannot rule out additional targets in rings, repeating all experiments during this parasite stage is beyond the scope of this study.

Question 3. Figure 2. Although Figure 2 is insightful and somehow self-explanatory, I think it misses two specific pieces of information. First, it is indicated in line 618 (M&M) that parasite material for thermal stability and limited proteolysis studies correspond to synchronized parasites (30-38h post-invasion) but this information is not given in Figure 2. In addition, if I fully understand the experimental protocol of obtaining parasite extracts, they strictly correspond to the soluble protein fraction of the erythrocytic stages of plasmodium at the late trophozoite stage, and not to all parasitic proteins as the scheme of Figure 2 might suggest. I would appreciate it very much if these two points (parasite stages and soluble proteins) were clearly indicated in the scheme as indeed, not the whole parasite blood stage proteome is investigated in the study but just a part of it (~47%, as the authors indeed indicate line 406). Please, edit also the legend of the figure accordingly.

This is correct, the soluble protein fraction from synchronised trophozoites was used in our proteomics studies. These details have been included in an updated Figure 2 and in the corresponding figure legend.

Question 4. Thermal stabilization. Figure 3B. Could the authors explain how they calculated or measured "absolute" protein abundances, and how this refers to a number of parasites in initial assays as this is not clear to me. Notably, abundance for PfA-M1 is much higher than for PF3D7_0604300, which are interesting "absolute" values.

Protein abundance was calculated using the mean peptide quantity of the stripped peptide sequence, with only precursors passing the Q-value threshold (0.01) considered for relative quantification. Within independent experiments, normalisation was based on total protein amount (determined by the BCA assay) rather than the initial number of parasites.

PfA-M1 is known to be a highly abundant protein and PF3D7_0604300 (as well as the other protein hits identified by thermal stability proteomics) are likely less abundant. It is noted that abundance is also dependent on ionisation efficiency and trypsin digestion efficiency. Therefore, we avoid comparing absolute abundances across proteins and use relative differences across conditions instead.

NB: the word “absolute” in the text (“absolute fold-change”) refers to the absolute value of the fold-change (i.e. positive or negative), and not to absolute quantification of proteins. The preceding text in each case clarifies that these are based on “relative peptide abundance”.

Question 5. Figure 5A. How do the authors explain peptides whose abundances are decreasing instead of increasing? Figure 5C. Could the authors provide digital cues (aa numbers or positions) on the ribbon representation of the PfA-M1 sequence? It is difficult to correlate the position of the 3D domains with respect to the primary structure of the protein. Also, the "yellow" supposed to show the "drug ligand" is really not very visible.

LiP-MS is based on the principle that ligand binding alters the local proteolytic susceptibility of a protein to a non-specific protease (in this case proteinase K, PK). In this sense, in LiP-MS we are not looking at variations in the stability of whole proteins (as is the case with thermal stability proteomics, where proteins detected with significantly higher abundance in treated relative to control samples reflects thermal stabilisation of the target due to ligand binding), but differences in peptide patterns between treated and control samples that reflect a change in the ability of PK to cleave the target. Thus, in the bound state, the ligand prevents proteolysis with PK. This results in decreased abundance of peptides with non-tryptic ends (as PK cannot access the region around where the ligand is bound) and increased abundance of the corresponding fully tryptic peptide, when compared to the free target. This concept is demonstrated in Fig. 4A and is explained in the text (lines 279-282) and Fig. 4 figure legend.

To aid visualisation, we have not added amino acid positions on the PfA-M1 sequence in Fig. 5, but have provided amino acid positions for all peptides in Supplementary File 3. We have also changed the colour of the ligand in Fig. 5C to blue and increased transparency of the binding and centre of mass neighbourhoods.

Question 6. Gametocyte assays. Line 824 states that several compounds were used as positive controls for anti-gametocyte activity (chloroquine, artesunate, pyronaridine, pyrimethamine, dihydroartemisinin, and methylene blue) and line 821 states that the biological effects are measured against puromycin. This is not very clear to me, could the authors comment on this?

This wording has been clarified in the methods to reflect that 5 µM puromycin was used as the positive control to calculate percent viability, whereas the other antimalarials were run in parallel as reference compounds with known anti-gametocyte activity (line 862).

Question 7. Metabolomics. Metabolomic assays were done on parasites at 28h pi, incubated for 1h with 3x EC50 of MIPS2673. You mention applying the drug on 2x10E8 infected red blood cells (line 838) but you do not explain how you isolate these infected red blood cells from non-infected red blood cells. Could you please specify this?

Metabolomics studies were performed such that cultures at 2% haematocrit and 6% trophozoite-stage parasitaemia (representing 2 x 108 cells in total, rather than 2 x 108 infected cells) were treated with compound or vehicle and after 1 h metabolites were extracted. This methodological detail has been clarified in the methods (line 875).

Question 8. Figure 3B. Does this diagram come from the experimental 3D structure created by the authors (8SLO) or from molecular modeling? Please specify in the legend (line 1305).

The diagram showing the binding mode of MIPS2673 bound to PfA-M1 comes from the experimentally determined 3D structure (PDB ID: 8SLO). This has now been stated in the figure legend. Note that the structural diagram refers to Fig. 1B (not Fig. 3B as indicated by the reviewer). The experimentally determined PfA-M1 structure with MIPS2673 bound (PDB ID: 8SLO) was also used to map LiP peptides and estimate the MIPS2673 binding site in Fig. 5, which is also now reflected in the appropriate section of the text (line 308) and Fig. 5 legend.

Question 9. Line 745. Why not indicate µm concentration for this H-Leu-NHMec substrate while it is indicated for the other substrates mentioned in the rest of the paragraph (H-Ala-NHMec, 20 μM, etc..). Also in this section (Enzyme assays) the pH at which the various enzymatic assays were done is missing.

All enzyme assays were performed at pH 8.0. The concentration of H-Leu-NHMec varied depending on the enzyme assayed, as follows: 20 µM for PfA-M1, 40 µM for PvA-M1 and 100 µM for ERAP1 and ERAP2. This information is now clearly stated in the methods section (lines 782 and 787) and as a footnote for Supplemental Table S1.

Question 10. Line 830, please define FBS.

Fetal bovine serum (FBS) has been added where appropriate (line 867).

Question 11. The authors mention in the title the targeting of several plasmodium species, but the only experimental study on the Plasmodium vivax species concerns the use of the recombinant enzyme Pv-M1. Authors also mention "multi-stage targets", but ultimately only look at erythrocyte stages and three different gametocyte stages.

We have now removed the words “cross-species” and “multi-stage” from the manuscript title and abstract so as not to overstate these findings. We have also added the word “potential” in the manuscript text to clarify that selective M1 inhibition could offer a potential multistage and cross species strategy for malaria.

Question 12. Supplemental Table S1. I would suggest replacing "Percent inhibition by MIPS2673 of PfA-M1 and Pv-M1 aminopeptidases compared to selected human M1 homologues" with "Percent inhibition by MIPS2673 of PfA-M1 and Pv-M1 aminopeptidase activities compared to selected human M1 homologues".

Done.

Question 13. Supplemental Table S3. Here you indicate IC50 while in text and Figure 1 you quote EC50. Why this difference?

This has now been changed to EC50 in Supplemental Table S3.

**Reviewer #2 (Recommendations For The Authors):**
Amendments that I would recommend in order to improve the presentation include all four parts of the study:(1) In vitro antiparasitic activity of MIPS2673.The authors showed that MIPS2673 inhibits parasite growth with IC50 of 324nM measured by a standard drug sensitivity assay, Fig 1C. This is all well and good, but it would be helpful to include at least one if not more other compounds such as antimalaria drugs and/or their earlier inhibitors (e.g. inhibitor 1) for comparisons. This is typically done to show that the assay in this manuscript is fully compatible with previous studies. It will also give a better view of how the selective inhibition of PfM1 kills the parasite, specifically.

Alongside MIPS2673, we also analysed the potency of the known antimalarial artesunate, which was found to have an EC50 of 4 nM. This value agrees with the expected potency of artesunate and indicates our MIPS2673 value of 324 nM is indeed compatible with previous studies. We have now reported the artesunate EC50 value for reference (lines 197-198 and Fig. S1).

Next, the authors proceeded to investigate the stage-specific effect of MIPS2673 but this time doing a survival assay instead of proper IC50 estimations Figure 1. I wonder why? Drug survival assays have typically very limited information content and measuring proper IC50 in stage-specific wash-off assays would be much more informative.

We performed single concentration stage specificity assays to determine the parasite asexual stage at which MIPS2673 is most active. This involved washing off the compound after a 24 h exposure in rings or trophozoites and determining parasite viability in the next asexual lifecycle. While a full dose response curve would allow generation of an EC50 value against the respective parasite stages, this information is unlikely to change the interpretation that MIPS2673 is more active against trophozoites stages than against rings.

Finally, in Figure 1E, the authors present the fact that the MIPS2673 arrests the parasite development. This is done by presenting a single (presumably representative) cell per time point. This is in my view highly insufficient. I recommend this figure be supplemented by parasite stage counts or other more comprehensive data representation. Also, the authors mention that while there is a growth arrest, hemoglobin is still being made. From the cell images, I can not see anything that supports this statement.

We thank the reviewer for this constructive comment and they are correct in their assessment that these are representative parasite images at the respective time points. To address the reviewers concerns we have now provided cell counts from each treatment condition (Fig. 1E) at selected time points, which shows parasite stalling at the ring to trophozoite transition under drug treatment. On reflection, we agree that it is difficult to determine the presence of haemozoin from our images and have removed this statement.

(2) Protein thermal shift profiling. In the next step, the authors proceed to carry out cellular thermal shift profiling to show that PfM1 indeed interacts with MIPS2673, this time in the context of the total protein lysates from *P. falciparum*. This section of the study is in my view quite solid and indeed it is nice to see that the inhibitor causes a thermal shift of PfM1 which further supports what was already expected: interaction.I have no problem with this study in terms of the technical outcome but I would urge the authors to tone down the interpretation of these results in two ways.Four other proteins were found to be shifted by the inhibitor which also indicates interactions. Calling it simply "off-target" interactions might not represent the truth. The authors should explore and in some way comment that interactions with these proteins could contribute to the MIPS2673 MOA. I do not suggest conducting any more studies but simply acknowledge this situation. Identifying more than one target is indeed very common in CETSA studies and it would be helpful to acknowledge this here as well.

We agree that identifying binding proteins in addition to the “expected” target is commonplace, and is indeed one of the benefits of this unbiased and proteome-wide approach. In the results and discussion, we have now amended our language to refer to these additional hits as MIPS2673-interacting proteins. In our original manuscript we dedicate a paragraph in the discussion to these additional interacting proteins and the likelihood of them being targets that contribute to antimalarial activity. Of these four additional interacting proteins, only the putative AP2 domain transcription factor (PF3D7_1239200) is predicted to be essential for blood stage growth and is therefore the only protein from this additional four that would likely contribute to antimalarial activity. These points are explicitly stated in the discussion (lines 530-550). Notably, all of the other interacting proteins identified in our thermal stability dataset were detected in our LiP-MS experiment but were not identified as interacting proteins by this method. The remaining three proteins were two non-essential *P. falciparum* proteins with unknown functions (PF3D7_1026000 and PF3D7_0604300) that are poorly described in the literature and a human protein (RAB39A). Further analysis of these other thermal stability proteomics hits in our LiP-MS dataset (see responses to Reviewer #3) identified none or only 1 significant LiP peptide from these proteins across our LiP-MS datasets, indicating they are likely to be false positive hits. Caveats around identifying protein targets by different deconvolution methods are also now addressed (lines 545-550).

At some point, the author argues that causing shifts of only four/five proteins including PfM1 shows that MIPS2673 does not interact with other (off) targets. Here one must be careful to present the lack of shifts in the CETSA as proof of no interaction. There are many reasons why thermal shifts are not observed including the physical properties of the individual proteins, detection limit etc. Again I suggest adjusting these statements accordingly.

We thank the reviewer for raising this important point and have now included additional discussion around this comment (lines 545-550).

Finally, I am not convinced that Figure 2 presents nothing more than the overall experimental scheme with not much new information. Many of such schemes were published previously in the original publication of thermal profiling. I would suggest omitting it from the main text and shifting it into supplementary methods etc.

We agree that similar schemes have been published previously, especially for thermal proteome profiling, and acknowledge the reviewer’s suggestion of moving this figure to the supplemental material. However, we have kept Fig. 2 in the main text as this scheme also incorporates a LiP-MS workflow for malaria drug target deconvolution (the first to do so) and also to satisfy the additional details requested for this figure by Reviewer #1 (question 3).

(3) Identification of MIPS2673 target proteins using LiP-MS. In the next step, the authors carried out the limited proteolysis analysis with the rationale that protein peptides that are near the inhibitor binding site will exhibit higher resilience to proteolysis. The authors did a very good job of showing this for PfM1-MISP2673 interaction. This part is very impressive from a technological perspective, and I congratulate the authors on such achievement. I imagine these types of studies require very precise optimizations and performance.Here, however, I struggle with the meaning of this experiment for the overall flow of the manuscript. It seems that the binding pocket of MIPS2673 is less known since the inhibitor was designed for it. In fact, the authors mentioned that the crystal structure of PfM1 is available. From this perspective, the LiP-MS study represents more of a technical proof of concept for future drug target analysis but has limited contribution to the already quite well-established PfM1-MISP2673 interaction. Perhaps this could be presented in this way in the text.

We thank the reviewer for this comment and they are correct that we solved the crystal structure of PfA-M1 bound to MIPS2673. We wish to highlight that the primary reason for performing the LiP-MS study was as an independent and complementary target deconvolution method to narrow down the shortlist of targets identified with thermal stability proteomics, and validate with high confidence that PfA-M1 is indeed the primary target of MIPS2673 in parasites. The use of a complementary approach based on a different biophysical principle (proteolytic susceptibility vs thermal stability) would also allow us to identify MIPS2673 interacting proteins that may not be detectable by thermal stability proteomics, for example targets that do not alter their thermal stability upon ligand binding. The text in the results and discussion has been amended to clarify these points (lines 266-268 and 545-550).

Furthermore, we agree that correctly predicting the MIPS2673 binding site on PfA-M1 using our LiP-MS peptide data is a technical proof of concept. Indeed, we wished to highlight the potential utility of LiP-MS for identifying both the protein targets of drugs and predicting their binding site, which is not possible with many other target deconvolution approaches. This point has been updated in the text (lines 303-304, 459-461).

(4) Metabolomic profiling of MIPS2673 inhibition showed a massive accumulation of short peptides which clearly indicates that this inhibitor blocks some proteolytic activity of short peptides, presumably products of upstream proteolytic activities. Here the authors argue, that because many of these detected short (di-/tri-) peptides could be mapped on the hemoglobin protein sequence, this must be their origin. Although this might be the case the author could not exclude the fact that at least some of these come from other sources (e.g. Plasmodium proteins). It would be quite helpful to comment on such a possibility as well. In particular, it was mentioned that the main subcellular localization of PfM1 is in the cytoplasm while most if not all hemoglobin digestion occurs in the digestive vacuole...?

Indeed, we agree that _Pf_A-M1 is likely processing both Hb and non-Hb peptides and do not definitively conclude that all dysregulated peptides must be derived from haemoglobin. A subset of dysregulated peptides cannot be mapped to haemoglobin and must have an alternative source such as other host proteins or turnover of parasite proteins. We have amended the discussion to better reflect these possible alternate peptide sources (480-482). Although the peptides detected in the metabolomics study (2-5 amino acids) are too short to be definitively assigned to any specific parasite or RBC protein, it is important to note that our analysis strongly indicates that the majority, but not all, of dysregulated peptides are more likely to originate from haemoglobin than other human or parasite proteins. This is based on sequence mapping, which was aided by acquiring MS/MS data for a subset of dysregulated peptides from which we derive accurate sequences (as opposed to residue composition inferred from total peptide mass) to more directly link dysregulated peptides to haemoglobin. We further quantified the sequence similarity of dysregulated peptides to all detectable proteins in the *P. falciparum* infected erythrocyte proteome (~4700 proteins), showing that these peptides are statistically more similar to haemoglobin than other host or parasite proteins.

The apparent disconnect between PfA-M1 localisation (cytosol) and the predominant site of haemoglobin digestion (digestive vacuole, DV) is explained by the fact that peptides originating from digestion of haemoglobin in the DV are required to be transported into the cytoplasm for further cleavage by peptidases, including PfA-M1. This point has now been clarified in the discussion (lines 473-474).

**Reviewer #3 (Recommendations For The Authors):**
(1) Thermal stability studies confirmed that PfA-M1 was a binding target, however, there were other proteins consistently identified in the thermal stability studies. This raises the question as to their potential role as additional targets of this inhibitor. The authors dismiss these because they are not metalloproteases, but further analysis is warranted. This is particularly important as the authors were not able to generate mutants using in vitro evolution of resistance strategies. This often indicates that the inhibitor has more than one target.

We thank the reviewer for this comment. The possibility of other targets contributing to MIPS2673 activity was also raised by Reviewer #2 (question 2) and is addressed above. Further to our response to Reviewer #2, we agree that the inability to generate resistant parasites in vitro could indicate that inhibition of multiple essential parasite proteins (including PfA-M1) contribute to MIPS2673 activity and do not rule out this possibility. It may also indicate the target has a very high barrier for resistance and is unable to tolerate resistance causing mutations as they are deleterious to function. Indeed, previous attempts to mutate PfA-M1 (references 12 and 50), and our own attempts to generate MIPS2673 resistant parasites in vitro (unpublished), were unsuccessful. It is important to note that of the hits reproducibly identified using thermal stability proteomics, only PfA-M1 and a putative AP2 domain transcription factor (PF3D7_1239200) are predicted to be essential for blood stage growth. We have explicitly stated that PF3D7_1239200 could also contribute to activity (line 533 and 537).

As we identified multiple hits with thermal stability proteomics we employed the complementary LiP-MS method to further investigate the target landscape of MIPS2673. PfA-M1 was the only protein reproducibly identified as the target through this approach. Importantly, the five proteins identified as hits by thermal stability proteomics were also detected in our LiP-MS datasets, but only PfA-M1 was identified as a target by both target deconvolution methods, strongly indicating it is the primary target of MIPS2673 in parasites. An important caveat is that we profiled the soluble proteome (we did not include detergents necessary for extracting membrane proteins as they may interfere with these stability assays) and other factors (e.g. the biophysical properties of the protein) will impact on whether ligand induced stabilisation events are detected. We have added additional text in the discussion around the above points (lines 545-550).

While we do not definitively rule out other MIPS2673 interacting proteins existing in parasites (that possibly also contribute to activity), our metabolomics studies indicated no functional impact by MIPS2673 outside of elevated levels of short peptides. This is indicative of aminopeptidase inhibition and the profile of peptide accumulation was distinct from a known PfA-M17 inhibitor, and other antimalarials, further pointing to selective inhibition of the PfA-M1 enzyme by MIPS2673 being responsible for antimalarial activity.

(2) The next set of experiments focused on a limited proteolysis approach. Again several proteins were identified as interacting with MIPS2673 including metalloproteases. The authors go on to analyze the LiP-MS data to identify the peptide from PfA-M1 which putatively interacts with MIPS2673. The authors are clearly focused on PfA-M1 as the target, but a further analysis of the other proteins identified by this method would be warranted and would provide evidence to either support or refute the authors' conclusions.

As PfA-M1 was the only protein reproducibly identified as an interacting protein across both LiP-MS experiments (and by thermal stability proteomics) we focused our analysis on this protein. However, we agree that further analysis of the other putative interacting proteins would be valuable. Additional analysis was performed (see new figure S4) on the other interacting proteins identified by thermal stability proteomics and the other interacting proteins identified in LiP-MS experiment one, as no other proteins (apart from PfA-M1) were identified as hits in the second LiP-MS experiment (lines 314-318, 495-505, 740-762 and Fig. S4). Using the common peptides detected across both LiP-MS experiments we mapped significant LiP peptides to the structures of the other putative MIPS2673-interacting proteins, where a structure was available and significant LiP-MS peptides were detected, and measured the minimum distance to expected binding sites. It is noted that when using the same criteria for a significant LiP peptide that we used for our PfA-M1 analysis, only one significant LiP peptide is identified from these other putative interacting proteins (YSPSFMSFK from PfADA). Therefore, we used a less stringent criteria for defining significant LiP peptides for these other proteins (see methods and Fig. S4 legend) in order to identify significant LiP peptides to map to structures. This analysis showed that, with the exception of PfA-M17, significant LiP-MS peptides for these other proteins are not significantly closer to binding sites than all other detected peptides, supporting our assertion that these other proteins are likely to be false positives or not functionally relevant MIPS2673 interacting proteins. Although significant peptides from PfA-M17 were closer to the binding site, our thermal stability and metabolomics data, combined with our previous work on the PfA-M17 enzyme, argue against this being a functionally relevant target (see lines 362-374 and 486-529 for a more detailed discussion). Another possible explanation for this result is that peptide substrates accumulating due to primary inhibition of PfA-M1 interact with PfA-M17, leading to structural changes around the enzyme active site that are detected by LiP-MS.

(3) The final set of experiments was an untargeted metabolomics analysis. They identified 97 peptides as significantly dysregulated after MIPS2673 treatment of infected cells and most of these peptides were derived from one of the hemoglobin chains. The accumulation of peptides was consistent with a block in hemoglobin digestion. This experiment does reveal a potential functional confirmation, but questions remain as to specificity.

As indicated, the accumulation of short peptides identified by metabolomics suggests MIPS2673 perturbs aminopeptidase function. Many of these peptides (but not all) likely map to haemoglobin and are more haemoglobin-like than other proteins in the infected red blood cell proteome. An effect on a subset of non-haemoglobin peptides is also apparent and we have added this to our discussion (also refer to our response to question 4 from Reviewer #2). A direct comparison to our previous metabolomics analysis of a specific PfA-M17 inhibitor (MIPS2571, reference 11) revealed MIPS2673 induces a unique metabolomic profile. The extent of peptide accumulation differed and a subset of short basic peptides (containing Lys or Arg) were elevated only by MIPS2673, consistent with the broad substrate preference of PfA-M1. Importantly, the metabolomics profile induced by MIPS2673 is the opposite of many other antimalarials, which cause depletion of haemoglobin peptides. Taken together, the profile of short peptide accumulation induced by MIPS2673 is consistent with specific inhibition of PfA-M1.

(4) Overall, this is an interesting series of experiments that have identified a putative inhibitor of PfA-M1 and PvA-M1. The work would be significantly strengthened by structure-aided analysis. It is unclear why putative binding sites cannot be analyzed via specific mutagenesis of the recombinant enzyme.

Contrary to this comment we solved the crystal structure of PfA-M1 bound to MIPS2673, determining its binding mechanism to the enzyme. This was further supported through proteomics-based structural analysis by LiP-MS. Undertaking site specific mutagenesis would be interesting to further probe the binding dynamics of MIPS2673 to the M1 protein. However, we believe it is beyond the scope of this study and would not change our conclusion that MIPS2673 binds to PfA-M1, which we have shown using multiple unbiased proteomics-based methods, enzyme assays and X-ray crystallography.

(5) In the thermal stability and LiP -MS analysis, other proteins were consistently identified in addition to PfA-M1 and yet no additional analysis was undertaken to explore these as potential targets.

As addressed in our previous responses, across independent thermal stability proteomics experiments we consistently identified 5 interacting proteins, including the expected target PfA-M1. In contrast, only PfA-M1 was reproducible across independent LiP-MS experiments. While several plausible putative targets (including aminopeptidases and metalloproteins) were identified in one of our LiP-MS experiment, they appear to be false discoveries and not responsible for the antiparasitic activity of MIPS2673, as peptide-level stabilisation was not consistent across independent LiP-MS experiments, and an interaction is refuted by our thermal stability, metabolomics and recombinant enzyme inhibition data. We have now performed further analysis of these other putative interacting proteins, which also argues against them being likely interacting proteins (see also response to question 2). We have also added to our existing discussion on possible MIPS2673 targets and the likelihood of these proteins contributing to antimalarial activity (lines 486-550).

(6) The metabolomics experiments were potentially interesting, but without significant additional work including different lengths of treatment and different stages of the parasite, the conclusions drawn are overstated. Many treatments disrupt hemoglobin digestion - either directly or indirectly and from the data presented here it is premature to conclude that treatment with MIPS2673 directly inhibits hemoglobin digestion.

Our metabolomics studies were performed using typical experimental conditions for investigating the antimalarial mechanisms of compounds by metabolomics (see references 11, 39, 40 and 55-57). We used a short 1 h incubation at 3x EC50 allowing us to profile the primary parasite pathways affected by MIPS2673 and avoid a nonspecific death phenotype associated with longer incubations. As addressed in our response to Reviewer #1 (question 2) we focused on trophozoite infected red blood cells as this is the stage most susceptible to MIPS2673 and when one presumes the greatest functional impact would be seen. It is possible that an expanded kinetic metabolomics analysis may reveal secondary mechanisms involved in MIPS2673 activity and we have now acknowledged this in the manuscript (lines 515-516). However, even though secondary mechanisms may become apparent at longer incubations it also becomes difficult to uncouple drug specific responses from nonspecific death effects. We believe any additional information provided by an expanded metabolomics analysis is unlikely to outweigh the significant extra financial cost associated with this type of experiment.

It is correct that many antimalarial compounds appear to disrupt haemoglobin digestion when analysed by metabolomics. However, as indicated in our manuscript (lines 369-373) and previous responses, the profile of elevated haemoglobin peptides induced by MIPS2673 is substantially different to the profile caused by other antimalarials. For example, artemisinins and mefloquine cause haemoglobin peptide depletion (references 55-57) and chloroquine results in increased levels of a different subset of non-haemoglobin peptides (see Creek et al. 2016). While there is some overlap in profile with a selective M17 inhibitor (our previous work, reference 11), the level of enrichment of these peptides is different and MIPS2673 also induces accumulation of a distinct set of basic peptides consistent with the substrate preference of the PfA-M1 enzyme. As we show that MIPS2673 does not inhibit other parasite aminopeptidases, a likely explanation for the profile overlap is that the build-up of substrates that cannot be processed by PfA-M1 leads to secondary dysregulation of other aminopeptidases. Our analyses (sequence mapping, MS/MS analysis and sequence similarities to all infected red blood cell proteins) strongly indicate that the majority of elevated peptides (but not all) originate from haemoglobin. Combined with our proteomics and recombinant enzyme data indicating direct engagement of PfA-M1, and with previous literature indicating the enzyme functions to cleave amino acids from haemoglobin-derived peptides, our data indicates MIPS2673 likely directly perturbs the haemoglobin digestion pathway through PfA-M1 inhibition.

(7) Finally, the potency of this compound on parasites grown in vitro is 300 nM - this would need improvements in potency and demonstration of in vivo efficacy in the SCID mouse model to consider this a candidate for a drug.

We do not propose MIPS2673 as an antimalarial candidate. The experiments presented here were centred on target validation rather than identification of an antimalarial lead, which may be the focus of future studies. To avoid this confusion, we have amended the manuscript title and language throughout to clarify this point.